# MASKED SKILL TOKEN TRAINING FOR HIERARCHICAL OFF-DYNAMICS TRANSFER

**Zeyu Feng[1], Haiyan Yin[1], Yew-Soon Ong[1,3], Harold Soh[2,4]**
[1]CFAR and IHPC, Agency for Science, Technology and Research (A*STAR), Singapore
[2]Department of Computer Science, School of Computing, National University of Singapore, Singapore
[3]Nanyang Technological University (NTU), Singapore
[4]Smart Systems Institute, National University of Singapore, Singapore
`{feng_zeyu, yin_haiyan}@a-star.edu.sg`
`asysong@ntu.edu.sg`
`harold@comp.nus.edu.sg`

## ABSTRACT

Generalizing policies across environments with altered dynamics remains a key challenge in reinforcement learning, particularly in offline settings where direct interaction or fine-tuning is impractical. We introduce **Masked Skill Token Training (MSTT)**, a fully offline hierarchical RL framework that enables policy transfer using observation-only demonstrations. MSTT constructs a discrete skill space via unsupervised trajectory tokenization and trains a skill-conditioned value function using masked Bellman updates, which simulate dynamics shifts by selectively disabling skills. A diffusion-based trajectory generator, paired with feasibility-based filtering, enables the agent to execute valid, temporally extended actions without requiring action labels or access to the target environment. Our results in both discrete and continuous domains demonstrate the potential of mask-guided planning for robust generalization under dynamics shifts. To our knowledge, MSTT is the first work to explore masking as a mechanism for simulating and generalizing across off-dynamics environments. It marks a promising step toward scalable, structure-aware transfer and opens avenues to explore multi-goal conditioning, and extensions to more complex, real-world scenarios.

## 1 INTRODUCTION

Reinforcement learning (RL) agents often struggle to generalize when deployed in environments that differ from their training conditions (Pinto et al., 2017; Clavera et al., 2019; Abdolshah et al., 2021; Hansen et al., 2021; Feng et al., 2023). In many practical scenarios involving navigation and manipulation, deployment environments exhibit structural changes (e.g., blocked passages or new obstacles) that invalidate parts of the agent's learned behavior repertoire. The changes do not typically require new tasks or capabilities, but render some previously executable behaviors infeasible. A natural question is: *can agents adapt to such dynamics shifts with minimal additional supervision, i.e., without extensive interaction or full retraining?*

In this work, we address this question in a practical setting where the agent is trained offline in a source environment and deployed zero-shot in a target environment with altered dynamics. At deployment, the agent receives a single *observation-only demonstration*—without action annotations—that illustrates feasible behavior in the new environment. These demonstrations may come from human demonstrators or be inferred from video, and are far cheaper to obtain than full action-labeled trajectories or interactive feedback. This setting enables lightweight, low-overhead adaptation and is well suited to real-world applications where interaction is expensive or infeasible.

We introduce **Masked Skill Token Training** (MSTT), a hierarchical RL framework for adaptation to feasibility constraints using *only offline data*. The key insight is that dynamics shifts can be abstracted as constraints over a learned space of temporally extended skills. MSTT learns these discrete skills from source trajectories using a vector-quantized variational autoencoder (VQ-VAE) (van den Oord et al., 2017; Mentzer et al., 2024), resulting in a compact and reusable skill vocabulary. During training, MSTT simulates environment changes by randomly masking out subsets of skills, and learns

a *feasibility-conditioned critic* via a masked Bellman operator that propagates value only through feasible skill sequences.

At test time, MSTT infers a binary skill mask from a single observation-only demonstration in the target environment, and plans using the learned critic under the inferred constraints. Unlike prior off-dynamics RL approaches such as DARC (Eysenbach et al., 2021) and VGDF (Xu et al., 2023a), which require action-labeled data or target interaction, MSTT operates fully offline and uses only observational signals to guide transfer. In contrast to standard hierarchical RL methods (Bagaria & Konidaris, 2020; Qiao et al., 2025), which assume that learned skills remain valid during deployment, MSTT explicitly models and adapts to shifts in skill feasibility. MSTT uses a diffusion-based trajectory generator trained on source data. At deployment, it samples candidate trajectories and discards those whose encoded skill tokens violate the inferred mask. This enables behavior synthesis aligned with test-time constraints, without needing to explicitly condition the generative model on the mask or retraining it for each deployment scenario.

We evaluate MSTT on continuous domains, specifically the Maze2D in D4RL, FetchReach and Habitat environments with test-time modifications and requirements. MSTT significantly outperforms offline RL and transfer RL baselines, demonstrating robust adaptation using only a single observation-only demonstration per environment. These results indicate the promise of MSTT as a step toward enabling structure-aware transfer in dynamic environments with minimal supervision. To our knowledge, this work is the first to tackle the challenging setting of skill-level adaptation under structural dynamics shifts using only observation-only demonstrations and fully offline training.

To summarize, our main contributions are:

- We formulate a new off-dynamics transfer setting where agents adapt using only observation-only demonstrations, without access to target actions or environment interaction.

- We propose *Masked Skill Token Training* (MSTT), a hierarchical RL framework that models dynamics-induced skill constraints using binary masks, and learns a feasibility-conditioned critic via masked Bellman updates.

- We integrate a diffusion-based trajectory generator to enable robust skill execution under test-time constraints, and demonstrate strong generalization across structurally altered environments.

## 2 RELATED WORKS

**Policy Transfer under Dynamics Mismatch.** Generalizing RL policies under dynamics mismatch is a key challenge for real-world deployment. DARC (Eysenbach et al., 2021) addresses Off-Dynamics RL (ODRL) (Lyu et al., 2024) by learning reward corrections using discriminators trained on transition dynamics. VGDF (Xu et al., 2023a) filters transferable transitions using model ensembles, while H2O (Niu et al., 2022) reweights offline datasets to better match target distributions. Domain randomization (Peng et al., 2018) trains robust policies via diverse simulated dynamics, but often requires extensive environment access. In the fully offline setting, DARA (Liu et al., 2022) modifies source rewards based on classifier-estimated transition alignment between source and target domains. BOSA (Liu et al., 2024) proposes supported value optimization to better mitigate out-of-distribution transitions. IGDF (Wen et al., 2024) filters source domain data using contrastive cross-domain representation learning on states, action and rewards. xDIO (Raychaudhuri et al., 2021) imitate skills across different morphologies. Offline Meta-RL methods (Nam et al., 2022; Ni et al., 2023; Xu et al., 2022; 2023b) tackle this problem by meta-learning on multiple source tasks with different dynamics and rewards. For example, PDT (Xu et al., 2022) achieves task transfer by training decision transformer on inputs that contain both a snippet of task demonstration and offline trajectory. HDP (Xu et al., 2023b) transfers adapter layers' parameters in decision transformer by learning on multiple source tasks with task demonstration for each task as a conditioning factor, generalizing to new task demonstration. Unlike these approaches, this work aims to achieve generalization across structural dynamics shifts without target interaction or explicit modeling of transition mismatches.

**Hierarchical Skills and Discrete Latent Policies.** Hierarchical RL (HRL) facilitates long-horizon decision-making by structuring policies around temporally extended skills, improving exploration, credit assignment, and policy reuse. While classical HRL relies on predefined subgoals or options, recent works learn skill abstractions from data, often through discrete latent policies that represent reusable primitives as compact codes. OPAL (Ajay et al., 2021) learns a discrete skill dictionary using

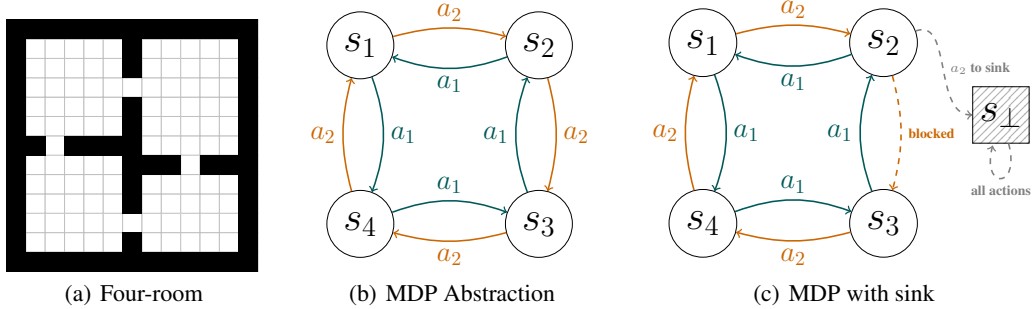

Figure 1: The Four-room environment and its skill-level MDP abstraction. (a) The original Four-room grid world, where the agent must reach specified goal rooms from varying start locations. (b) Abstract MDP representation, where rooms are states and inter-room transitions are treated as high-level skills $a_i$. (c) Modified MDP with a blocked transition (e.g., the door from room 2 to room 3 is impassable), where executing the corresponding skill leads to a sink state. MSTT learns a feasibility-conditioned critic through simulated skill masking, enabling transfer to such target domains without interaction.

VQ-VAE (van den Oord et al., 2017; Mentzer et al., 2024; Ozair et al., 2021), and QueST (Mete et al., 2024) encodes variable-length trajectory tokens for multitask reuse. DADS (Sharma et al., 2020) and VALOR (Achiam et al., 2018) focus on skill predictability and diversity to enhance generalization. SiMPL (Nam et al., 2022) learns skill embedding and a high-level policy over the skill conditioned on task embedding. Li et al. (2021; 2023) propose theoretically grounded and sample-efficient ways to learn options and embeddings. In particular, HiT-MDP (Li et al., 2023) theoretically proves that option-induced SMDP is homomorphic equivalents to MDP with option embeddings, greatly improving sample efficiency and convergence stability of policy optimization in the corresponding MDP. Furthermore, VMOC (Li et al., 2025) extends this idea to the use of latent thoughts as options in an abstract reasoning space for efficient and effective reasoning. These approaches offer modularity and efficiency but implicitly assume that skills remain executable during deployment. Our method removes this assumption by introducing a skill-masking mechanism that enables policies to reason over partially-available skills.

**Diffusion-Based Generative Policy Learning.** Diffusion models have emerged as an expressive policy class for offline RL (Kang et al., 2023). Diffuser (Janner et al., 2022) formulates planning as trajectory denoising through conditional diffusion. Decision Diffuser (Ajay et al., 2023) extends this idea to decision-making, modeling return-conditioned trajectories. Diffusion Policies (Chi et al., 2023) apply diffusion to visuomotor control, producing pixel-to-action policies with high fidelity. Latent Diffusion RL (Venkatraman et al., 2024) leverages low-dimensional latent spaces to improve efficiency in trajectory generation. LTLDoG (Feng et al., 2024) and DOPPLER (Feng et al., 2025) generates novel trajectories and actions to satisfy symbolic temporal logic specifications with diffusion models. These methods achieve strong results in static environments, capable of composing hierarchical offline knowledge (Zhou et al., 2023; Du et al., 2023; Liang et al., 2024; Ma et al., 2024), but assume fixed dynamics and do not explicitly handle feasibility constraints introduced by changes in environment structure. MetaDiffuser (Ni et al., 2023) trains a context encoder from trajectories with transition and reward information from different tasks. Our method augments this line of work by conditioning diffusion-based skill generation on masked skill availability, enabling policy rollout under dynamics-induced constraints.

## 3 SKILL-LEVEL MDP ABSTRACTION WITH MASKED FEASIBILITY

Recall that our goal is to enable decision-making under dynamics shifts, where certain skills may become infeasible due to changes in transition feasibility (e.g., due to physical obstacles or altered connectivity[1]). In this section, we formalize the connection between skill feasibility and transition dynamics using a skill-level abstraction of the MDP.

---

[1]In this work we primarily consider structural dynamics changes such as locked passages or new obstacles, not continuous changes on parameters of transition, e.g. mass, friction or damping. Transfer across continuous changes requires training on trajectories from randomly sampled parameters, and we pursue this as future work.

We introduce the notion of a *skill mask*, a binary indicator over the discrete skill space, which captures environment-specific feasibility constraints. This mask provides a compact mechanism to encode target environment dynamics at the skill level, and allows us to analyze how such feasibility constraints affect value estimation and policy behavior. Our formulation supports off-dynamics policy transfer by explicitly modeling which skills remain executable in the target environment, without requiring direct interaction to capture its underlying dynamics or action-labeled trajectories.

To illustrate this abstraction, we use the Four-room navigation domain as an example of a discrete hierarchical reinforcement learning (Sutton et al., 1999) task involving structured decision-making with reusable behavioral primitives. As shown in Fig. 1(a), the environment comprises four interconnected rooms with doors at fixed positions. The agent is tasked with navigating from different start locations to designated goal rooms using previously learned skills that enable room-to-room transitions. These skills can be modeled as *temporally extended actions*.

**Skill Abstraction of the Source MDP.**   We assume that a set of temporally extended skills are available, which enables the agent to transition between high-level regions of the environment (e.g., adjacent rooms). Based on these behaviors, we construct a skill-level abstraction of the environment in the form of a discrete augmented action-space Markov Decision Process (MDP) (Puterman, 2014) $\mathcal{M} := (\mathcal{S}, \mathcal{A}, r, p, \gamma)$, where $\mathcal{S}$ denotes the abstract state space (e.g., room indices), $\mathcal{A}$ denotes the set of skill-level actions, $r$ is the reward function defined over abstract states, $p$ captures the skill-induced transition dynamics, and $\gamma$ is the discount factor . This high-level abstraction is illustrated in Fig. 1(b), and serves as the foundation for reasoning over skill composition in our framework.

**Skill-Level Feasibility Masking.**   In deployment settings, environment dynamics may differ from those observed during training, rendering certain skill-level transitions infeasible (e.g., a blocked passage between room 2 and room 3). To capture such structural constraints, we augment the abstract state space with a dedicated sink state $s_\perp$, which absorbs any transition resulting from the execution of an invalid skill (Fig. 1c). This sink state formulation provides a conservative but practical model of failure, where executing an unavailable skill results in termination—this reflects scenarios such as safety violations or irreversible errors. While our current formulation adopts this absorbing failure model, alternative semantics (e.g., remaining in the same state upon executing an invalid skill) are also possible and are left for future work.

The resulting target MDP is denoted $\mathcal{M}_{\boldsymbol{B}} := (\mathcal{S} \cup \{s_\perp\}, \mathcal{A}, r, p_{\boldsymbol{B}}, \gamma)$, where $\boldsymbol{B} \in \{0,1\}^{|\mathcal{S}| \times |\mathcal{S}|}$ is a blocking matrix indicating the feasibility of transitions. $B_{i,j} = 0$ denotes that the transition from $s_i$ to $s_j$ is blocked under the current dynamics:

$$p_{\boldsymbol{B}}(s_j|s_i, a) := 0 \ \text{if} \ B_{i,j} = 0; \qquad p_{\boldsymbol{B}}(s_\perp|s_i, a) := \sum_{s_j \in \mathcal{S}} (1 - B_{i,j})p(s_j|s_i, a)$$

where transitions to blocked states are redirected to a sink state $s_\perp$, which models failure due to infeasible skill execution. We also introduce a binary availability mask $m(s,a) \in \{0,1\}^{|\mathcal{S}||\mathcal{A}|}$, where $m(s,a) = 0$ denotes that action $a$ is blocked at state $s$.

**Masked Bellman Operator.**   To support learning in the presence of unknown blocked dynamics, we introduce a masked Bellman operator that uses the binary availability mask $m(s,a) \in \{0,1\}^{|\mathcal{S}||\mathcal{A}|}$, where $m(s,a) = 0$ denotes that action $a$ is blocked at state $s$. The masked Bellman operator modifies the standard backup by restricting the value propagation to feasible transitions:

$$(\mathcal{T}^m Q)(s, a) := r(s) + \gamma \langle m(s,a)p(s,a), \mathcal{V}_Q^m \rangle, \tag{1}$$

where $\mathcal{V}_Q^m(s)$ is a masked value operator that computes the *maximum* over available actions:

$$\mathcal{V}_Q^m(s) := \begin{cases} \max\limits_{a \in \mathcal{A}:m(s,a)=1} Q(s,a), & \text{if such } a \text{ exists} \\ r(s), & \text{otherwise.} \end{cases} \tag{2}$$

In other words, this masked operator enables the critic to account for feasibility constraints during learning by preventing value propagation through blocked or invalid transitions.

**Theoretical Justification.**   Under blocked dynamics, iterative updates using the masked Bellman operator can converge to a near-optimal policy. Specifically, for any target MDP with infeasible transitions, there exists an availability mask that enables convergence to a policy whose value closely approximates that of the optimal policy under the modified dynamics:

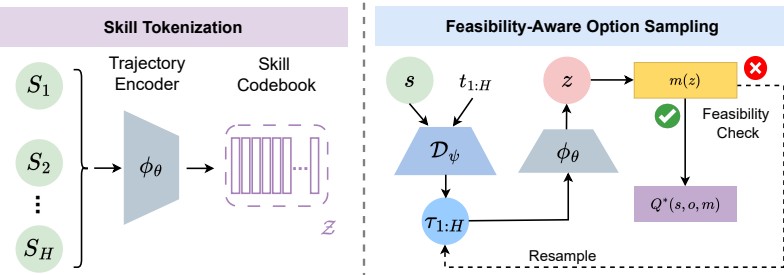

Figure 2: Overview of our masked skill token learning framework. (Left) Skill tokenization encodes trajectory segments into discrete latent tokens via a VQ-VAE encoder $\phi_\theta$. (Right) At execution time, the diffusion model $\mathcal{D}_\psi$ samples candidate sub-trajectories, which are encoded into skill tokens ($z$). Feasibility is enforced via the binary mask $m(z)$; only valid skills are propagated for Q-learning.

**Theorem 1.** *Let $Q_0 : \mathcal{S} \times \mathcal{A} \to \mathbb{R}_+$ be an initial action-value function. There exists a mask $m$ such that the iterative update $Q_{k+1} = \mathcal{T}^m Q_k$ converges to a policy $\pi_m^K$ whose value under the blocked MDP $\mathcal{M}_B$ satisfies:*

$$\left\| V_{\mathcal{M}_B}^{\pi_m^K} - V_{\mathcal{M}_B}^* \right\|_\infty \le \alpha \gamma^K + \beta(1 - p_{\min}), \tag{3}$$

*where $p_{\min} := \min_{s,a} \max_{s'} p(s'|s, a)$ and $\alpha, \beta$ are constants dependent on $\gamma$.*

For more detailed derivation and proof, please see Appendix A.

### 3.1 VALIDATING MASKED BELLMAN UPDATES

We empirically validate Theorem 1 using the example discrete environment shown in Figure 1(a). We simulate masked updates in the unblocked source MDP (Figure 1(b)) using a feasibility mask where action 2 is unavailable at state $s_2$. The resulting value function $V_K$ is computed through $K$ iterations of the masked Bellman operator. We evaluate performance across different levels of $p_{\min}$ (which reflects the entropy of the transition dynamics).

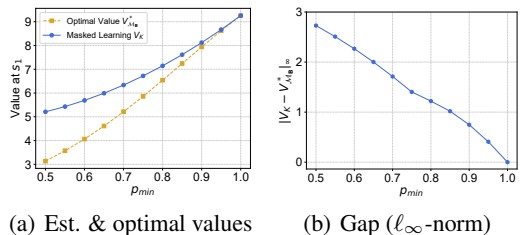

(a) Est. & optimal values   (b) Gap ($\ell_\infty$-norm)

Figure 3compares the value function estimated via the masked Bellman operator to the analytically derived optimal values under blocked dynamics. Notably, in low-entropy regimes where $p_{\min} \approx 1$, the largest gap (Figure 3(b)) between estimated and optimal values becomes negligible. These findings empirically validate our theoretical findings and highlight the potential of our approach in structured settings where skill transitions are predictable.

Figure 3: Performance of the masked Bellman operator using Eq. (1) and Eq. (2). (a) Converged value estimates (Est.) at state $s_1$ compared with the analytically computed optimal values under blocked dynamics. (b) $\ell_\infty$-norm of the difference between the estimated and optimal value functions. Both plots show that the masked Bellman operator produces accurate estimates, particularly as $p_{\min}$ approaches 1.

In summary, Theorem 1 shows that masked value learning via the Bellman operator $\mathcal{T}^m$ can approximate optimal behavior in a target MDP $\mathcal{M}_B$, provided a suitable feasibility mask $m$ is applied. In hierarchical settings, where skills are temporally extended and induce low-entropy transitions, the approximation error remains small (e.g., when $p_{\min} \approx 1$). This property allows dynamics shifts to be simulated in the source domain by masking skill availability during training. As a result, value functions can be learned entirely offline and still generalize to test environments without requiring access to their true dynamics. We leverage this insight to design the training framework described in the next section.

---

**Algorithm 1** Offline Masked Hierarchical Reinforcement Learning

---

**Require:** Trajectory dataset $D$, total training steps $E$, batch size $B$
 1: Initialize critics $Q_{\varphi_1}, Q_{\varphi_2}$ and targets $Q_{\varphi_1'}, Q_{\varphi_2'}$
 2: **for** $e = 0$ to $E$ **do**
 3:    Sample $\{\boldsymbol{\tau}_{1:H}^{(b)}\}_{b \in [B]}$ uniformly randomly from $D$
 4:    Sample $\{m^{(b)}\}_{b \in [B]}$ randomly from $\{0, 1\}^L$
 5:    Sample $N$ target skills at $s_H^{(b)}$ and $m^{(b)}$ using Algorithm 2
 6:    Select best skill $o'^{(b)}$ by $Q_{\varphi_1'}$
 7:    Get clipped Gaussian noised version $\tilde{o}'^{(b)}$
 8:    Get Bellman loss $\ell_j^{(b)}$ from Eq. (5) with double Q-learning
 9:    Update $\varphi_j \leftarrow \varphi_j - \eta \nabla_{\varphi_j} \frac{1}{B} \sum_b \ell_j^{(b)}, j \in \{1, 2\}$
10:    **if** $e \bmod e_0 == 0$ **then**
11:        $\varphi_j' = \lambda \varphi_j + (1 - \lambda) \varphi_j', j \in \{1, 2\}$
12:    **end if**
13: **end for**
14: **return** $Q_{\varphi_1}$

---

# 4 MASKED SKILL TOKEN LEARNING FOR OFF-DYNAMICS RL

In this section, we transition to learning in continuous state and action spaces, where the discrete skill abstractions assumed in Section 3 are not directly available.

We introduce *Masked Skill Token Training* (MSTT), a hierarchical RL framework that learns a compact skill vocabulary from offline data and supports policy transfer under altered dynamics. MSTT comprises three main components: (1) it encodes sub-trajectories into discrete skill tokens using a VQ-VAE trained on offline demonstrations; (2) it learns a feasibility-aware critic using masked Bellman updates that simulate skill infeasibility through random masking; and (3) it uses a diffusion-based policy to generate candidate skills, filtering them according to an inferred skill feasibility mask obtained from a single observation-only demonstration in the target environment. We now describe each component in detail.

## 4.1 SKILL TOKENIZATION VIA UNSUPERVISED TRAJECTORY ENCODING

In continuous environments where the state-action space cannot be directly abstracted into a finite MDP, we obtain reusable skill representations by learning discrete encodings of raw trajectory segments from offline data. Specifically, we adapt the VQ-VAE to discover latent behavioral primitives (sub-trajectories) that form a compact symbolic skill vocabulary.

Let $\boldsymbol{\tau}_{1:H} = (s_1, \ldots, s_H)$ denote a sub-trajectory with horizon $H$, where we let $s_h$ denote an observation with slight abuse of notation. We encode this trajectory using the VQ-VAE encoder $\phi_\theta$ to produce a discrete latent token:

$$z = \phi_\theta(\boldsymbol{\tau}_{1:H}), \quad z \in \mathcal{Z} := \{1, 2, \ldots, L\},$$

where $L$ is the size of the learned codebook $\mathcal{Z}$. Each token $z \in \mathcal{Z}$ indexes a cluster of similar sub-trajectories in the dataset and thus, the codebook $\mathcal{Z}$ provides a discrete abstraction over the continuous behavior space, with each token capturing consistent transition patterns. We find that even when trained with observation-only sub-trajectories, the learned codebook captures meaningful spatial and behavioral structure in the environment.

The VQ-VAE model is trained by minimizing the following loss function over sub-trajectories:

$$\mathcal{L}_{\text{skill-enc}} = \underbrace{\|\boldsymbol{\tau}_{1:H} - \xi_\vartheta(z)\|^2}_{\text{reconstruction}} + \underbrace{\|\text{sg}[\phi_\theta(\boldsymbol{\tau}_{1:H})] - z\|^2}_{\text{codebook loss}} + \beta \underbrace{\|\phi_\theta(\boldsymbol{\tau}_{1:H}) - \text{sg}[z]\|^2}_{\text{commitment}}, \tag{4}$$

where $\xi_\vartheta$ is the decoder, $\text{sg}[\cdot]$ denotes the stop-gradient operator, and $\beta$ is a hyperparameter that balances codebook commitment against the other loss terms.

## 4.2 DIFFUSION-BASED OFF-DYNAMICS SKILL ADAPTATION

To enable hierarchical decision-making under dynamics shifts, we consider each discrete skill token $z \in \mathcal{Z}$ as a temporally extended action in an augmented action-space MDP. Each skill $o$ is instantiated using a trajectory segment $(s_1, a_1, \ldots, s_H)$ and the internal policy of the skill corresponds to the action sequence. Our formulation of the skill as fixed-horizon temporally extended actions can be viewed as a simplified version of the options framework (Sutton et al., 1999; Sutton & Barto, 2018), where option $o = \langle \mathcal{I}, \pi_o, \mathcal{B} \rangle$, $\mathcal{I} \subseteq \mathcal{S}$ is the initiation set, $\pi_o$ is the intra-option policy, and $\mathcal{B} : \mathcal{S} \to [0, 1]$ is the termination condition. In our case, the skill $o$ terminates deterministically after $H$ steps.

To support planning, we extend the masked Bellman operator (Eq. (1)) to operate over skills:

$$Q^*(s, o, m) = R(o) + \gamma^H \max_{o' \in \mathcal{O}(s') \,:\, m(\phi_\theta(o'))=1} Q^*(s', o', m), \qquad (5)$$

where $\mathcal{O}(s')$ is the set of valid skills at the successor state $s'$. The return $R(o)$ is computed as the discounted sum of per-step rewards. If no feasible skill is available, the value defaults to the terminal reward as in Eq. 2.

We apply the VQ-VAE trajectory encoder $\phi_\theta$ (Section 4.1) to associate each skill with a skill token by encoding the sub-trajectory of observations $z = \phi_\theta(\boldsymbol{\tau}_{1:H})$. Following the abstraction introduced in Section 3, we define a skill feasibility mask $m(z)$, which models which skills remain executable under altered dynamics: i.e., $m(z) = 1$ indicates a feasible skill, while $m(z) = 0$ denotes an infeasible one.

We sample skills using a trajectory-level diffusion policy model $\mathcal{D}_\psi$, trained to generate plausible observation-action sub-trajectories. Unlike the VQ-VAE decoder, the diffusion model supports conditioning on continuous start states $s$,

---

**Algorithm 2** Off-Dynamics Skill Sampling

**Require:** Start $s$, Diffusion model $\mathcal{D}_\psi$, encoder $\phi_\theta$, mask $m$
1: $z \leftarrow \texttt{null}$
2: **while** $z ==$ $\texttt{null}$ **do**
3:      Sample $\boldsymbol{\tau}_{1:H} \sim \mathcal{D}_\psi(\cdot \mid s)$
4:      $z \leftarrow \phi_\theta(\boldsymbol{\tau}_{1:H})$
5:      **if** $m(z) == 0$ **then**
6:          $z \leftarrow \texttt{null}$      ▷ Reject and resample
7:      **end if**
8: **end while**
9: **return** Feasible skill $\boldsymbol{\tau}_{1:H}$ and $z$

---

enabling flexible skill generation in diverse environments. Rather than conditioning the model directly on the feasibility mask, we apply a simple filtering procedure: sampled trajectories whose encoded skill tokens are masked out are discarded (Algorithm 2). The accepted trajectory is then executed as a temporally extended skill. This mechanism allows for zero-shot composition of valid high-level behaviors under novel dynamics, guided solely by the feasibility mask.

## 4.3 MODEL TRAINING AND INFERENCE

The overall training procedure is summarized in Algorithm 1. At each iteration, we randomly sample trajectory segments and skill masks from the offline dataset, and update the Q-function by minimizing the squared Bellman error under the masked update rule (Eq. (5)). Although the number of possible skill-mask combinations grows combinatorially with the size of the skill vocabulary, we find that the learned skill space is sufficiently structured and random masking provides effective coverage in practice. To stabilize training, we incorporate several standard techniques from deep Q-learning, including target networks (Mnih et al., 2016), clipped double Q-learning (Hasselt, 2010; Fujimoto et al., 2018), and target smoothing via noise injection (Simmons-Edler et al., 2019) similar to DOPPLER (Feng et al., 2025).

At test time, MSTT infers the feasibility mask from a single observation-only demonstration. The demonstration can be non-expert as long as it covers a path towards the goal. MSTT is capable of compositing skills present in the demonstration to achieve the goal. We begin by initializing the skill mask $m(z) = 0$ for all $z \in \mathcal{Z}$. Then, we extract all sub-trajectories $\boldsymbol{\tau}_{1:H}$ from the demonstration and encode them using the trajectory encoder to obtain skill tokens $z = \phi_\theta(\boldsymbol{\tau}_{1:H})$. For each observed skill token, we set $m(z) = 1$, marking the corresponding skill as feasible. This inferred mask is passed to the learned critic and used to guide sampling via Algorithm 2.

## 5 EXPERIMENTS

We evaluate MSTT's ability to achieve observation-only demonstration following in previously unseen environments with altered dynamics, without additional training or fine-tuning. Our primary

Table 1: Performance on Maze2D test suites.

| Env | Metric | BC | Diffuser | Diffuser$_t$ | BC$_{ta}$ | DARA | MSTT (ours) |
|---|---|---|---|---|---|---|---|
| T1 | Return (↑) | 0.0±0.0 | 83.5±50.78 | 107.36±27.41 | 57.76±18.24 | 39.13±31.93 | 163.97±67.11 |
| | Steps (↓) | 350.0±0.0 | 280.65±42.17 | 162.84±27.02 | 157.49±15.61 | 310.87±31.93 | 186.92±66.85 |
| | Goal (↑) | 0% | 73% | 99% | 99% | 65% | 89% |
| T2 | Return (↑) | 0.19±1.89 | 15.86±39.31 | 36.05±42.49 | 19.32±7.11 | 63.67±22.20 | 145.51±50.93 |
| | Steps (↓) | 349.8±1.98 | 336.7±32.96 | 298.54±66.76 | 187.37±57.37 | 286.33±22.20 | 211.01±46.74 |
| | Goal (↑) | 1% | 14% | 44% | 93% | 93% | 95% |
| T3 | Return (↑) | 8.44±12.84 | 0.0±0.0 | 27.94±22.04 | 22.94±21.73 | 61.83±37.60 | 111.59±34.74 |
| | Steps (↓) | 310.85±57.89 | 350.0±0.0 | 259.44±63.34 | 280.29±64.85 | 288.17±37.60 | 249.33±35.02 |
| | Goal (↑) | 32% | 0% | 73% | 54% | 78% | 99% |
| Avg | Return (↑) | 2.87 | 33.12 | 57.11 | 33.34 | 54.87 | 140.35 |
| | Steps (↓) | 336.88 | 322.45 | 240.27 | 208.38 | 295.12 | 215.75 |
| | Goal (↑) | 11% | 29% | 72% | 82% | 78.66% | 94.33% |

$t$: fine-tuned on target state-only demonstrations. $ta$: fine-tuned on target action demonstrations.

Table 2: FetchReach test results.

| Metric | BC | Diffuser | Diffusert$_t$ | BC$_{ta}$ | DARA | MSTT (ours) |
|---|---|---|---|---|---|---|
| Return (↑) | -50.0±0.0 | -46.95±3.12 | -30.85±11.44 | -50.0±0.0 | -49.64±0.75 | -37.26±7.43 |
| Cost (↓) | 0.0±0.0 | 200.0±206.51 | 112.0±99.02 | 0.0±0.0 | 208±85.65 | 0.0±0.0 |
| Goal (↑) | 0% | 99% | 50% | 0% | 23% | 88% |
| Failure (↓) | 0% | 61% | 66% | 0% | 93% | 0% |

$t$: fine-tuned on target state-only demonstrations. $ta$: fine-tuned on target action demonstrations.

objective is to benchmark MSTT against state-of-the-art offline and transfer RL methods in popular simulation environments (Figures 4, 5 and 7). The empirical study includes visualizations of skill masks (see also appendix for more details) and generalization results to provide insights into MSTT's mask sampling behavior. We also include a case study showcasing the use of a Vision-Language Model (VLM) to infer demonstrations from visual observations.

## 5.1 EXPERIMENTAL SETUP

**Environments.** We evaluate MSTT in three simulated environments from Gymnasium-Robotics suite (de Lazcano et al., 2024) and Habitat-Lab (Savva et al., 2019; Szot et al., 2021; Puig et al., 2024): Maze2D and FetchReach, commonly used in offline (Janner et al., 2022) and hierarchical RL (Shin & Kim, 2023) research, and a photorealistic 3D indoor navigation task in ReplicaCAD (Szot et al., 2021) commonly used in embodied AI. Maze2D (Figure 4(a)) requires long-horizon navigation with continuous control and sparse rewards (1 at the goal). Learning is difficult from non-expert, fully offline data without hierarchical modeling. At test time, we block certain paths (Eysenbach et al., 2021), creating target environments where the original policy fails. We evaluate on three such variants (Figures 4(b)–4(d)), each allowing only a single feasible route. The agent receives a single locations-only demonstration per environment and must infer the required skills to follow it to the

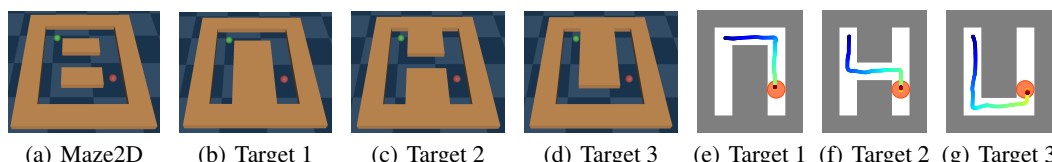

| (a) Maze2D | (b) Target 1 | (c) Target 2 | (d) Target 3 | (e) Target 1 | (f) Target 2 | (g) Target 3 |

Figure 4: Environments and test-time performance of MSTT in Maze2D. (a) Source environment with continuous state and action spaces. Three distinct long-horizon paths connect the start location (green ball) to the goal (red ball), each requiring hundreds of low-level control steps. (b)-(d) Test-time variants with altered dynamics, where only one path remains feasible. The agent must navigate to the goal given observation-only demonstration. (e)-(g) Trajectories generated by MSTT in target domains. The agents successfully finished navigation by selecting the correct path.

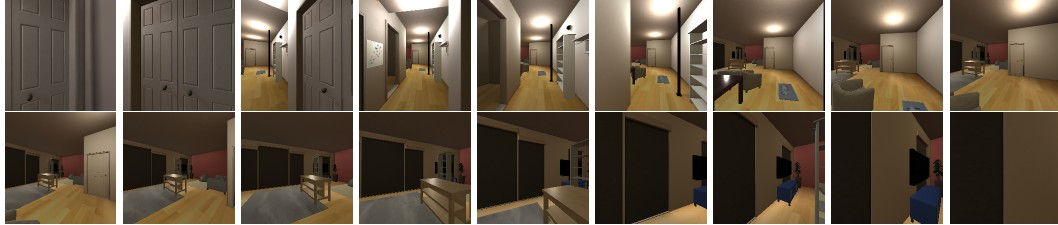

Figure 5: Demonstration in Habitat ReplicaCAD. The agent must navigate to the goal in this studio with pixel observations, following the demonstration images in order from left to right, then top to bottom.

Table 3: Habitat ReplicaCAD test results. Cost measures the average deviation of the trajectory coordinates from the demonstrated trajectory.

| Metric | Diffuser | Diffuser$_t$ | ID | MSTT |
|---|---|---|---|---|
| Cost ($\downarrow$) | 2.4±3.7 | 2.2±1.9 | 1.8±1.5 | 0.7±0.6 |
| Goal ($\uparrow$) | 10% | 16% | 23% | 75% |

$t$: fine-tuned on target state-only demonstrations.

Table 4: MSTT test results with VLM generated state demonstration.

| Metric | Maze T1 | Maze T2 | Maze T3 | FetchReach |
|---|---|---|---|---|
| Return ($\uparrow$) | 188.6±47.9 | 152.2±42.2 | 111.5±35.6 | -35.0±6.44 |
| Steps ($\downarrow$) | 162.3±47.6 | 198.7±42.1 | 239.4±35.5 | 35.0±6.4 |
| Goal ($\uparrow$) | 94% | 97% | 98% | 93% |
| Failure ($\downarrow$) | 6% | 3% | 2% | 0% |

goal. In FetchReach (Figure 7(a)), the robot controls a gripper to reach target positions, receiving a reward of $-1$ per step. In the target variant (Figure 7(b)), a red cuboid defines a constrained region that incurs a heavy penalty if entered. The agent must imitate a safe coordinates-only demonstration to reach the goal while avoiding the penalty zone. In ReplicaCAD (Figure 5), the agent must navigate to a goal in a photorealistic indoor environment using only RGB observations, following a sequence of images as a demonstration.

**Compared Methods and Evaluation Metrics.** We compare MSTT against several popular of-fline RL and imitation learning baselines: Diffuser (Janner et al., 2022) and Behavior Cloning (BC) (Pomerleau, 1988; Levine et al., 2020), both of which learn from offline data but lack built-in mechanisms for adapting to new environments. To enable adaptation, we fine-tune Diffuser on target observation-only demonstrations, denoted as Diffuser$_t$. We also include DARA (Liu et al., 2022), an offline dynamics-aware transfer RL method that extends DARC (Eysenbach et al., 2021) to the offline setting using action-annotated target data. Additionally, BC$_{ta}$ denotes behavior cloning fine-tuned with action-labeled target demonstrations. Importantly, MSTT uses only state-only demonstrations in the test environment, while DARA and BC$_{ta}$ rely on action annotations. Although these methods have more information about the target environment, we include their results for completeness and comparison. Another relevant method that can transfer with observation-only demonstrations is inverse dynamics (ID) modeling (Radosavovic et al., 2021). However, we find that it performs poorly in the continuous control tasks in Maze2D and FetchReach, due to the difficulty of accurately learning the actions from locations alone without robot velocities. Therefore, we only include it in the Habitat experiment. Evaluation metrics include total episode return, number of steps to reach the goal, and success/failure rates over 100 trials. We report the mean and standard deviation across trials for each target task.

## 5.2 RESULTS

**Can MSTT achieve transfer in the off-dynamics scenario with only observation-only demon-strations?** MSTT achieves near-expert performance in all Maze2D variants, FetchReach, and ReplicaCAD, outperforming all baselines on average, as shown in Tables 1, 2 and 3. Offline RL methods, such as Diffuser and BC, fail to adapt to the changed dynamics and perform poorly in transfer settings. Fine-tuning Diffuser on observation-only demonstrations (Diffuser$_t$) improves performance, particularly in Target Env 1, where the model benefits from pretraining bias toward the correct path. However, it still lags behind MSTT overall. DARA performs inconsistently, showing moderate success in some settings (e.g., Target Env 2), but struggles to generalize; this is likely due to limited hierarchical abstraction and a poorly trained domain classifier affected by dataset imbalance. Planning with inverse dynamics models performs poorly in ReplicaCAD with overfitting to high-

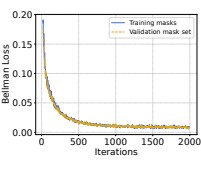 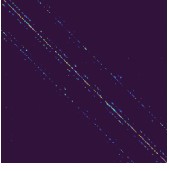

(a) Validation     (b) Skill transition

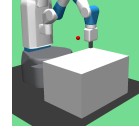 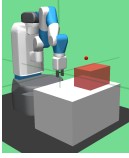 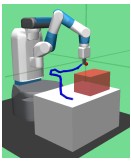 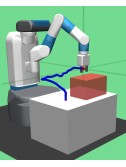

(a) Fetch     (b) Target     (c) Demo     (d) Result

Figure 6: Analysis of MSTT's training with random masks on skills in Maze2D. (a) Training loss on training masks and validation mask set. (b) The empirical skill indices transition matrix estimated from the dataset.

Figure 7: Environments and test-time performance of MSTT in Fetch. (a) The source environment, where the robot gripper must reach the goal location (red ball). (b) The target environment with modified dynamics: the red box indicates a restricted region that the gripper must avoid. (c) Demonstration, where the robot gripper is tasked with reaching the goal with avoidance. (d) MSTT successfully executes a trajectory that reaches the goal while avoiding the red box.

dimensional inputs. These results highlight MSTT's strength in leveraging skill-level abstractions and feasibility-aware planning for robust adaptation and demonstration following.

**How does MSTT learn a generalizable critic across different skills with random mask sampling?**
The critic learned in MSTT can generalize across different skill combinations, as evidenced by the training and validation loss curves in Figure 6(a). The training loss decreases steadily, while the validation loss on a fixed set of masks that are not used for sampling during training also decreases and remains close to the training loss, indicating good generalization.

While the skill library can grow large with a bigger codebook size, we find that the skills learned by VQ-VAE are often well-structured and the transitions between skills are sparse. Figure 6(b) (skill transition matrix) illustrates the transition matrix of the skill tokens extracted from the source dataset, revealing structured, banded patterns in skill encoding and usage: transitions concentrate along patterned indices, indicating temporally consistent and semantically related motion primitives. Therefore, many masking patterns become irrelevant due to the sparse transitions in training, and the random masking strategy provides sufficient coverage of the skill space during training, allowing the critic to generalize well at test time. More visualizations in Section C.2 in the appendix further demonstrates the semantic meanings of the learned skill embeddings. MSTT effectively identifies and samples skills relevant to the demonstrated trajectory, while masking out irrelevant ones.

**Case study: Using VLMs with MSTT.** We investigate the use of modern vision-language models (VLMs) to infer observation-only demonstrations directly from visual observations. This approach reduces annotation overhead and illustrates how foundation models can be used with MSTT for off-dynamics transfer. Specifically, we prompt VLM to generate coordinates sequences. Full prompt details, model outputs, and analysis are provided in the appendix. As shown in Table 4, MSTT achieves strong performance using these VLM-inferred demonstrations, demonstrating the potential of combining pretrained VLMs with our framework for low-effort generalization to new environments.

## 6 CONCLUSION AND FUTURE WORK

In this work, we introduced MSTT, an offline hierarchical RL framework that uses masked skill tokens and diffusion-based skills to enable skill transfer across off-dynamics environments. By conditioning value learning on binary skill masks, MSTT simulates skill infeasibility and avoids the need for action annotations or explicit dynamics modeling. Experiments in both discrete and continuous settings demonstrate robust transfer performance under dynamics shifts. Future work includes improving mask sampling strategies for efficiency, incorporating diverse dynamics parameters with domain randomization, extending to multi-goal transfer via goal-conditioned critics, and further leveraging LLMs or VLMs to infer demonstrations from visual input to reduce reliance on human supervision. We believe MSTT forms an important step towards scalable skill transfer in complex, real-world settings.

ACKNOWLEDGEMENTS

This research is supported by the National Research Foundation, Singapore and Infocomm Media Development Authority under its Trust Tech Funding Initiative, and the National Research Foundation, Singapore, under its Thematic Competitive Research Programme 2025 (NRF-T-CRP-2025-0003). This research is partly supported by the MTI under its AI Centre of Excellence for Manufacturing (AIMfg) (Award W25MCMF014). Any opinions, findings and conclusions or recommendations expressed in this material are those of the author(s) and do not reflect the views of National Research Foundation, Singapore and Infocomm Media Development Authority.

REPRODUCIBILITY STATEMENT

We have taken the following steps to enhance the reproducibility of our results. For our theoretical results we state required assumptions and problem setups in Section A.1, and include full proofs in Section A.2. Our simulated environmental setup, dataset processing and experimental pipelines are documented in Section B. We provide implementation details of the proposed algorithm and computing environments in Section B of the appendix.

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

APPENDIX

## A  THEORETICAL ANALYSIS

In this section, we present the theoretical results that support the claims made in the main text. We begin by outlining the detailed problem setup, followed by the statement of the main theorem. The proof of the theorem is provided thereafter.

### A.1  NOTATIONS AND PROBLEM FORMULATION

Let $\Delta^{S-1}$ denote the space of probability distributions over a discrete set $\mathcal{S}$ and $S = |\mathcal{S}|$. For a conditional probability mass function $p(Y|X)$, we denote the probability vector given $X = x$ as $\boldsymbol{p}(x) \in \Delta^{S-1}$. Let $\mathbb{R}_+ = \{x \in \mathbb{R} : x \geq 0\}$. The $L^\infty$-norm of a vector $\boldsymbol{v}$ is denoted as $\|\boldsymbol{v}\|_\infty := \max_i |[v]_i|$, where $[v]_i$ refers to the $i$-th entry of $\boldsymbol{v}$. The inner product between two vectors $\boldsymbol{u}$ and $\boldsymbol{v}$ is $\langle \boldsymbol{u}, \boldsymbol{v} \rangle$.

In this analysis we consider a finite Markov decision process (MDP) (Puterman, 2014) denoted as $\mathcal{M} := (\mathcal{S}, \mathcal{A}, r, p, \gamma)$, which consists of finite state space $\mathcal{S}$, finite action space $\mathcal{A}$, transition probability function $p \colon \mathcal{S} \times \mathcal{A} \to \Delta^{S-1}$, reward function $r \colon \mathcal{S} \to [0, 1]$ and discount factor $\gamma \in [0, 1)$. The fixed discounted factor here is motivated by our practical choice that skills represented as temporally extended actions in augmented action-space MDP with diffusion model have fixed horizon as shown in Section 4.2, making the original semi-MDP equivalent to an abstraction MDP. A policy $\pi \colon \mathcal{S} \to \Delta^{A-1}$ maps states to distributions over actions. An agent equipped with a policy can interact with the environment by taking action $a_t$ at time step $t$ according to $\pi$ based on the current state $s_t$, and observing the resulting state transitions and rewards. The environment transitions to a new state $s_{t+1}$ according to $\boldsymbol{p}(s_t, a_t)$ conditioned on the current state and action. The agent receives

a reward $r(s_t)$ based on the current state. Our analysis can be generalized to the reward function $r\colon \mathcal{S} \times \mathcal{A} \to [0,1]$ by modifying the masked Bellman operator below. But for sake of simplicity in analytical representation, we assume the reward function is state dependent only.

The agent's goal is to maximize the expected sum of discounted rewards over time, or expected return. The objective at $s$ is exactly the value function given by $V^\pi(s) = \mathbb{E}_{\pi,p}[R|s_0 = s] = \mathbb{E}_{\pi,p}[\sum_{t=0}^\infty \gamma^t r(s_t)|s_0 = s]$, where $R$ is the return for a trajectory and $\gamma$ is the discount factor that determines the importance of future rewards. The action-value function $Q^\pi(s,a) = \mathbb{E}_{\pi,p}[R|s_0 = s, a_0 = a]$ represents this expected return conditioned on a specific initial $s$ and $a$. We denote the vectors of a value function and an action-value function as $\boldsymbol{V}^\pi \in \mathbb{R}_+^S$ and $\boldsymbol{Q}^\pi \in \mathbb{R}_+^{SA}$, respectively. We define the operator $\mathcal{V}$ for a function $f\colon \mathcal{S} \times \mathcal{A} \to \mathbb{R}_+$ such that for each $s$, $(\mathcal{V}f)(s) := \max_{a \in \mathcal{A}} f(s,a)$. The Bellman operator for $f$ is denoted as $(\mathcal{T}f)(s,a) := r(s) + \gamma\langle \boldsymbol{p}(s,a), \mathcal{V}\boldsymbol{f}\rangle$. Here we also overload $\mathcal{V}\boldsymbol{f}$ and $\mathcal{T}\boldsymbol{f}$ to denote the corresponding vector obtained after applying the operators. Let $\pi^\star$ be the optimal policy and $Q^\star$ be the optimal Q function. The optimal Bellman equation for $Q^\star$ can be written as $\boldsymbol{Q}^\star = \mathcal{T}\boldsymbol{Q}^\star$, which lays the foundation for many reinforcement learning algorithms where $p$ and $r$ are unknown (Sutton & Barto, 2018) to the learner.

In this work, the target test environments may exhibit altered dynamics due to unavailability of certain transitions, *e.g.*, physical traps or obstructions that block previously accessible paths. For each transition, we can define a binary function $b\colon \mathcal{S} \times \mathcal{S} \to \{0,1\}$, where $b(s_i, s_j) = 1$ indicates that the transition from $s_i$ to $s_j$ is available, and unavailable otherwise. Its matrix form is denoted as $\boldsymbol{B} \in \{0,1\}^{S \times S}$, where $b_{i,j} := [B]_{i,j} = b(s_i, s_j)$. $b_{i,j} = 0$ indicates that the transition path from $s_i$ to $s_j$ is blocked, and any actions applied at $s_i$ will not lead to $s_j$. For example, in Figure 1 of the main text, this corresponds to the case where there is a one-way door from room 3 to room 2. When such transition attempts are made following the original MDP transition, the agent will collide with the obstruction and fail the episode, which is equivalent to having 0 rewards in the future.

Therefore, we can model a target environment as an MDP $\mathcal{M}_{\boldsymbol{B}} := (\mathcal{S} \cup \{s_\perp\}, \mathcal{A}, r, p_{\boldsymbol{B}}, \gamma)$ with same action space and reward, but different state space and transition probability function $p_{\boldsymbol{B}}\colon \mathcal{S} \cup \{s_\perp\} \times \mathcal{A} \to \Delta^S$. For all $b_{i,j} = 0$, we have $p_{\boldsymbol{B}}(s_j|s_i, a) = 0$ for all $a \in \mathcal{A}$, indicating that this transition is not possible. Since the agent fails the episode when it was about to have an unavailable transition, we can introduce a sink state $s_\perp$ where $r(s_\perp) = 0$ and $p_{\boldsymbol{B}}(s_\perp|s_\perp, a) = 1, \forall a \in \mathcal{A}$. The probability of transitioning to other states at $s_\perp$ is 0. The probability of transitioning to $s_\perp$ from $s_i$ for any $a$ is $p_{\boldsymbol{B}}(s_\perp|s_i, a) = \sum_{s_j \in \mathcal{S}}(1 - b_{i,j})p(s_j|s_i, a)$. $p_{\boldsymbol{B}}$ remains same as $p$ for all other transitions. The related values in $\mathcal{M}_{\boldsymbol{B}}$ are denoted as $\boldsymbol{V}_{\mathcal{M}_{\boldsymbol{B}}}, \boldsymbol{Q}_{\mathcal{M}_{\boldsymbol{B}}}$ and $\boldsymbol{V}_{\mathcal{M}_{\boldsymbol{B}}}^\star, \boldsymbol{Q}_{\mathcal{M}_{\boldsymbol{B}}}^\star$ for optimal policy.

During training on the source environment $\mathcal{M}$, we cannot efficiently modify the real dynamics to either learn from samples distributed as from the target environment, or learn for many possible target environments at the same time. We aim to rely on the trajectories collected from the source environment to approximately learn a policy for a target environment. One way to achieve this is directly modifying the transition tuple according to $\boldsymbol{B}$ during the Bellman update by changing the target state to a sink state if the intent transition is unavailable. However, it is difficult to scale to continuous state environments as it would require comparing state similarity with a suitable state space partition. Instead, here we tackle this problem based on $s$ and $a$ that lead to the unavailable transition. Our algorithm in practice deals with partial observations of the state. The formulation here can align with this practice by considering the aggregations of states with same observation as an abstract state.

Given $\boldsymbol{B}$, we define mask $m(s_i, a_i) = 0$ for any transition $(s_i, a_i, s_j)$ if $b_{i,j} = 0$ and $s_j \in \arg\max_{s \in \mathcal{S}} p(s|s_i, a_i)$, and $m(s_i, a_i) = 1$ otherwise. With this mask vector $\boldsymbol{m} \in \{0,1\}^{SA}$, we introduce the following masked Bellman operator

$$(\mathcal{T}^{\boldsymbol{m}}Q)(s,a) := r(s) + \gamma\langle m(s,a)\boldsymbol{p}(s,a), \mathcal{V}^{\boldsymbol{m}}\boldsymbol{Q}\rangle, \tag{6}$$

where

$$(\mathcal{V}^{\boldsymbol{m}}f)(s) := \begin{cases} \max_{a \in \mathcal{A}: m(s,a)=1} f(s,a) & \text{if } \exists a \in \mathcal{A}: m(s,a) = 1 \\ r(s) & \text{otherwise} \end{cases} \tag{7}$$

is a masked operator for a function $f\colon \mathcal{S} \times \mathcal{A} \to \mathbb{R}_+$. Eq. (6) is general in that it can be developed into a loss function based on this temporal difference for learning in continuous environment. Compared

with modifying the transition tuple according to $B$, $\mathcal{T}^m$ is more efficient as it does not require checking the next state in the transition tuple. Moreover, during deployment stage, $\mathcal{V}^m Q$ can filter out unavailable skills or actions based on the mask, whereas the $B$ approach would require learning a dynamics model to predict the next state for checking. Our objective here is to examine the difference of the proposed masked Bellman operator $\mathcal{T}^m$ compared with learning under the exact $B$ by theoretically analyzing its behavior and performance.

## A.2 THEOREM AND PROOF

We restate Theorem 1 below which now includes the detailed formulation.

**Theorem 1.** *Let $Q^0 \colon \mathcal{S} \times \mathcal{A} \to \mathbb{R}_+$ be the initial action-value function randomly initialized as $\left\| Q^0 \right\|_\infty \in \left[ 0, \frac{1}{1-\gamma} \right]$. The masked Bellman operator $\mathcal{T}^m$ iteratively updates $Q^k$ for $k = 0, 1, \dots$ by $Q^{k+1} = \mathcal{T}^m Q^k$. The performance of the masked greedy policy $\pi_m^K(s)$, which chooses action according to $\pi_m^K(s) := \arg\max_{a \in \mathcal{A} : m(s,a)=1} Q^K(s,a)$ if $\exists a \in \mathcal{A} : m(s,a) = 1$ and randomly otherwise, is close to the optimal value function $V_{\mathcal{M}_B}^\star$ in the target environment after $K$ iterations:*

$$\left\| V_{\mathcal{M}_B}^{\pi_m^K} - V_{\mathcal{M}_B}^\star \right\|_\infty \le \frac{2\gamma^K}{(1-\gamma)^2} + \frac{2\gamma(1+\gamma)(1-\gamma^K) + \gamma(1-\gamma)^2}{(1-\gamma)^3} (1 - p_{\min}), \qquad (3)$$

*where $p_{\min} := \min_{s \in \mathcal{S}, a \in \mathcal{A}} p(s_d(s,a)|s,a)$ and $s_d(s,a) \in \arg\max_{s' \in \mathcal{S}} p(s'|s,a)$.*

To proof this theorem we start with the following lemmas. Since the value function in $\mathcal{M}_B$ has an additional sink state $s_\perp$, we extend the definition of both $\mathcal{T}^m$ and $\mathcal{V}^m$ by letting $(\mathcal{T}^m f)(s_\perp, \cdot) := 0$ and $(\mathcal{V}^m f)(s_\perp) := \max_{a \in \mathcal{A}} f(s_\perp, a)$ for any function $f \colon \mathcal{S} \cup \{s_\perp\} \times \mathcal{A} \to \mathbb{R}_+$.

**Lemma 1.** $\left\| \mathcal{V}^m Q_{\mathcal{M}_B}^\star - \mathcal{V} Q_{\mathcal{M}_B}^\star \right\|_\infty \le \frac{\gamma}{1-\gamma} (1 - p_{\min})$.

*Proof.* Based on the definitions, for any $s \in \mathcal{S}$,

$$\left| \left( \mathcal{V}^m Q_{\mathcal{M}_B}^\star \right)(s) - \left( \mathcal{V} Q_{\mathcal{M}_B}^\star \right)(s) \right| \tag{LHS}$$

$$= \begin{cases} \left| \max_{a \in \mathcal{A} : m(s,a)=1} Q_{\mathcal{M}_B}^\star(s,a) - \max_{a \in \mathcal{A}} Q_{\mathcal{M}_B}^\star(s,a) \right| & \text{if } \exists a \in \mathcal{A} : m(s,a) = 1 \\ \left| r(s) - \max_{a \in \mathcal{A}} Q_{\mathcal{M}_B}^\star(s,a) \right| & \text{otherwise.} \end{cases} \tag{8}$$

Consider the first case when $\exists a \in \mathcal{A} : m(s,a) = 1$. We analyze two subcases based on the value of $m(s, \pi^\star(s))$. Let $\pi^\star(s) \in \arg\max_{a \in \mathcal{A}} Q_{\mathcal{M}_B}^\star(s,a)$, and the lemma is clearly true when $m(s, \pi^\star(s)) = 1$ as LHS $= 0$. In the other subcase where $m(s, \pi^\star(s)) = 0$, by letting $\pi_m^\star(s) \in \arg\max_{a \in \mathcal{A} : m(s,a)=1} Q_{\mathcal{M}_B}^\star(s,a)$, we have

$$\text{LHS} = \max_{a \in \mathcal{A}} Q_{\mathcal{M}_B}^\star(s,a) - \max_{a \in \mathcal{A} : m(s,a)=1} Q_{\mathcal{M}_B}^\star(s,a) \tag{9}$$

$$= r(s) + \gamma \left\langle p_B(s, \pi^\star(s)), \mathcal{V} Q_{\mathcal{M}_B}^\star \right\rangle - r(s) - \gamma \left\langle p_B(s, \pi_m^\star(s)), \mathcal{V} Q_{\mathcal{M}_B}^\star \right\rangle \tag{10}$$

$$\le \gamma \sum_{s' \in \mathcal{S} \cup \{s_\perp\}} p_B(s'|s, \pi^\star(s)) \max_{a' \in \mathcal{A}} Q_{\mathcal{M}_B}^\star(s', a') \tag{11}$$

$$\le \gamma \sum_{s' \in \mathcal{S}} p_B(s'|s, \pi^\star(s)) \frac{1}{1-\gamma} \tag{12}$$

$$= \frac{\gamma}{1-\gamma} \sum_{s' \in \mathcal{S} \setminus \{s_d(s, \pi^\star(s))\}} p_B(s'|s, \pi^\star(s)) \tag{13}$$

$$\le \frac{\gamma}{1-\gamma} \sum_{s' \in \mathcal{S} \setminus \{s_d(s, \pi^\star(s))\}} p(s'|s, \pi^\star(s)) \tag{14}$$

$$= \frac{\gamma}{1-\gamma} \left( 1 - p(s_d(s, \pi^\star(s)) | s, \pi^\star(s)) \right) \tag{15}$$

$$\le \frac{\gamma}{1-\gamma} (1 - p_{\min}). \tag{16}$$

In Eq. (11) the optimal action-value function at the sink state is 0. Eq. (13) is due to the fact that we are in the case $m(s, \pi^\star(s)) = 0$, where there exists a $s_d(s, \pi^\star)$ such that $b(s, s_d(s, \pi^\star)) = 0$.

In the other case when $\forall a \in \mathcal{A} : m(s,a) = 0$, we have

$$\text{LHS} = \left| r(s) - r(s) - \gamma \left\langle \boldsymbol{p_B}\left(s, \pi^\star(s)\right), \mathcal{V}\boldsymbol{Q}^\star_{\mathcal{M}_B} \right\rangle \right| \tag{17}$$

$$= \gamma \left| \sum_{s' \in \mathcal{S} \cup \{s_\perp\}} p_B\left(s'|s, \pi^\star(s)\right) \max_{a' \in \mathcal{A}} Q^\star_{\mathcal{M}_B}\left(s', a'\right) \right| \tag{18}$$

$$\leq \frac{\gamma}{1-\gamma}\left(1 - p_{\min}\right), \tag{19}$$

by following similar steps from Eq. (11).

The inequality holds at $s_\perp$ since the optimal $Q$ function is $0$ at $s_\perp$. The lemma is then proved by summarizing the above analysis under the $L^\infty$-norm. $\qquad \square$

**Lemma 2.** *The masked Bellman operator $\mathcal{T}^{\boldsymbol{m}}$ is a contraction mapping with respect to the $L^\infty$-norm, i.e., for any two vectors $\boldsymbol{f}$ and $\boldsymbol{g}$, we have $\|\mathcal{T}^{\boldsymbol{m}}\boldsymbol{f} - \mathcal{T}^{\boldsymbol{m}}\boldsymbol{g}\|_\infty \leq \gamma \|\boldsymbol{f} - \boldsymbol{g}\|_\infty$.*

*Proof.* To handle vectors in the space of $\mathbb{R}_+^{(S+1)A}$, similar to the extension of $\mathcal{T}^{\boldsymbol{m}}$, we also denote $p_e$ as an extended version of $p$ by letting $p_e\left(s_\perp|s, a\right) = 0$ for all $s \in \mathcal{S} \cup \{s_\perp\}$ and $a \in \mathcal{A}$. Based on Eq. (6) and Eq. (7), we have

$$\left| [\mathcal{T}^{\boldsymbol{m}}\boldsymbol{f} - \mathcal{T}^{\boldsymbol{m}}\boldsymbol{g}]_{s,a} \right| \tag{LHS}$$

$$= \left| r(s) + \gamma \left\langle m(s,a)\boldsymbol{p_e}\left(s, a\right), \mathcal{V}^{\boldsymbol{m}}\boldsymbol{f} \right\rangle - r(s) - \gamma \left\langle m(s,a)\boldsymbol{p_e}\left(s, a\right), \mathcal{V}^{\boldsymbol{m}}\boldsymbol{g} \right\rangle \right| \tag{20}$$

$$= \gamma \left| \left\langle m(s,a)\boldsymbol{p_e}(s,a), \mathcal{V}^{\boldsymbol{m}}\boldsymbol{f} - \mathcal{V}^{\boldsymbol{m}}\boldsymbol{g} \right\rangle \right| \tag{21}$$

$$= \gamma \left| \sum_{s' \in \mathcal{S}} m(s,a)p\left(s'|s, a\right)\left(\left(\mathcal{V}^{\boldsymbol{m}}f\right)\left(s'\right) - \left(\mathcal{V}^{\boldsymbol{m}}g\right)\left(s'\right)\right) \right|. \tag{22}$$

For any $s$, we can bound

$$\left| \left(\left(\mathcal{V}^{\boldsymbol{m}}f\right)(s) - \left(\mathcal{V}^{\boldsymbol{m}}g\right)(s)\right) \right| \tag{23}$$

$$= \begin{cases} \left| \max_{a \in \mathcal{A}: m(s,a)=1} f(s,a) - \max_{a \in \mathcal{A}: m(s,a)=1} g(s,a) \right| & \text{if } \exists a \in \mathcal{A} : m(s,a) = 1 \\ |r(s) - r(s)| & \text{otherwise} \end{cases} \tag{24}$$

$$\leq \begin{cases} \max_{a \in \mathcal{A}: m(s,a)=1} |f(s,a) - g(s,a)| & \text{if } \exists a \in \mathcal{A} : m(s,a) = 1 \\ 0 & \text{otherwise} \end{cases} \tag{25}$$

$$\leq \max_{a \in \mathcal{A}} |f(s,a) - g(s,a)|. \tag{26}$$

Therefore,

$$\text{LHS} \leq \gamma \sum_{s' \in \mathcal{S}} m(s,a)p\left(s'|s, a\right) \max_{a' \in \mathcal{A}} |f\left(s', a'\right) - g\left(s', a'\right)| \tag{27}$$

$$\leq \gamma \max_{s' \in \mathcal{S} \cup \{s_\perp\}, a' \in \mathcal{A}} |f\left(s', a'\right) - g\left(s', a'\right)| \tag{28}$$

$$= \gamma \|\boldsymbol{f} - \boldsymbol{g}\|_\infty. \tag{29}$$

$\square$

**Lemma 3.** *For any function $f : \mathcal{S} \times \mathcal{A} \to \mathbb{R}_+$, $\left\|\boldsymbol{V}^{\pi_{f,\boldsymbol{m}}}_{\mathcal{M}_B} - \boldsymbol{V}^\star_{\mathcal{M}_B}\right\|_\infty \leq \frac{2\left\|\boldsymbol{f} - \boldsymbol{Q}^\star_{\mathcal{M}_B}\right\|_\infty + \gamma(1 - p_{\min})}{1-\gamma}$.*

*Proof.* As stated in Theorem 1, $\pi_{f,\boldsymbol{m}}$ is defined as the greedy policy with respect to $\boldsymbol{f}$ and $\boldsymbol{m}$: $\pi_{f,\boldsymbol{m}} \in \arg\max_{a \in \mathcal{A}: m(s,a)=1} f(s,a)$ if $\exists a \in \mathcal{A} : m(s,a) = 1$ and randomly otherwise. We can analyze the difference in two cases based on the value of $m(s, \pi^\star(s))$. For any $s \in \mathcal{S}$, consider the case where $m\left(s, \pi^\star(s)\right) = 0$,

$$V^\star_{\mathcal{M}_B}(s) - V^{\pi_{f,\boldsymbol{m}}}_{\mathcal{M}_B}(s) \tag{30}$$

$$=r(s) + \gamma \left\langle \boldsymbol{p_B}(s, \pi^\star(s)), \mathcal{V}\boldsymbol{Q}^\star_{\mathcal{M}_B} \right\rangle - r(s) - \gamma \left\langle \boldsymbol{p_B}(s, \pi_{f,\boldsymbol{m}}(s)), V^{\pi_{f,\boldsymbol{m}}}_{\mathcal{M}_B} \right\rangle \tag{31}$$

$$\leq \gamma \sum_{s' \in \mathcal{S} \cup \{s_\perp\}} p_B(s'|s, \pi^\star(s)) \max_{a' \in \mathcal{A}} Q^\star_{\mathcal{M}_B}(s', a') \tag{32}$$

$$\leq \frac{\gamma}{1-\gamma}(1 - p_{\min}), \tag{33}$$

similar to the steps in Lemma 1. In the other case where $m(s, \pi^\star(s)) = 1$,

$$V^\star_{\mathcal{M}_B}(s) - V^{\pi_{f,\boldsymbol{m}}}_{\mathcal{M}_B}(s) \tag{34}$$

$$=Q^\star_{\mathcal{M}_B}(s, \pi^\star(s)) - Q^\star_{\mathcal{M}_B}(s, \pi_{f,\boldsymbol{m}}(s)) + Q^\star_{\mathcal{M}_B}(s, \pi_{f,\boldsymbol{m}}(s)) - Q^{\pi_{f,\boldsymbol{m}}}_{\mathcal{M}_B}(s, \pi_{f,\boldsymbol{m}}(s)) \tag{35}$$

$$\leq Q^\star_{\mathcal{M}_B}(s, \pi^\star(s)) - f(s, \pi^\star(s)) + f(s, \pi_{f,\boldsymbol{m}}(s)) - Q^\star_{\mathcal{M}_B}(s, \pi_{f,\boldsymbol{m}}(s))$$
$$+ \gamma \left\langle \boldsymbol{p_B}(s, \pi_{f,\boldsymbol{m}}(s)), \boldsymbol{V}^\star_{\mathcal{M}_B} - \boldsymbol{V}^{\pi_{f,\boldsymbol{m}}}_{\mathcal{M}_B} \right\rangle \tag{36}$$

$$\leq 2 \left\| \boldsymbol{f} - \boldsymbol{Q}^\star_{\mathcal{M}_B} \right\|_\infty + \gamma \left\| \boldsymbol{V}^\star_{\mathcal{M}_B} - \boldsymbol{V}^{\pi_{f,\boldsymbol{m}}}_{\mathcal{M}_B} \right\|_\infty. \tag{37}$$

Noting that the value function at the sink state is 0, arranging the above inequality gives $\left\| \boldsymbol{V}^{\pi_{f,\boldsymbol{m}}}_{\mathcal{M}_B} - \boldsymbol{V}^\star_{\mathcal{M}_B} \right\|_\infty \leq \frac{2\|\boldsymbol{f} - \boldsymbol{Q}^\star_{\mathcal{M}_B}\|_\infty}{1-\gamma}$. The lemma is proved by summarizing the above analysis under the $L^\infty$-norm. □

Now we are ready to prove Theorem 1.

*Proof of Theorem 1.* Similar to the extension of the operators $\mathcal{T}^{\boldsymbol{m}}$ and $\mathcal{V}^{\boldsymbol{m}}$, we also extend $\boldsymbol{Q}^k$ to be well-defined on $s_\perp$. Thus, by Lemma 2,

$$\left\| \boldsymbol{Q}^K - \boldsymbol{Q}^\star_{\mathcal{M}_B} \right\|_\infty = \left\| \mathcal{T}^{\boldsymbol{m}} \boldsymbol{Q}^{K-1} - \boldsymbol{Q}^\star_{\mathcal{M}_B} \right\|_\infty \tag{38}$$

$$= \left\| \mathcal{T}^{\boldsymbol{m}} \boldsymbol{Q}^{K-1} - \mathcal{T}^{\boldsymbol{m}} \boldsymbol{Q}^\star_{\mathcal{M}_B} + \mathcal{T}^{\boldsymbol{m}} \boldsymbol{Q}^\star_{\mathcal{M}_B} - \boldsymbol{Q}^\star_{\mathcal{M}_B} \right\|_\infty \tag{39}$$

$$\leq \gamma \left\| \boldsymbol{Q}^{K-1} - \boldsymbol{Q}^\star_{\mathcal{M}_B} \right\|_\infty + \left\| \mathcal{T}^{\boldsymbol{m}} \boldsymbol{Q}^\star_{\mathcal{M}_B} - \boldsymbol{Q}^\star_{\mathcal{M}_B} \right\|_\infty \tag{40}$$

$$\leq \gamma^2 \left\| \boldsymbol{Q}^{K-2} - \boldsymbol{Q}^\star_{\mathcal{M}_B} \right\|_\infty + (1 + \gamma) \left\| \mathcal{T}^{\boldsymbol{m}} \boldsymbol{Q}^\star_{\mathcal{M}_B} - \boldsymbol{Q}^\star_{\mathcal{M}_B} \right\|_\infty \tag{41}$$

$$\leq \cdots \tag{42}$$

$$\leq \gamma^K \left\| \boldsymbol{Q}^0 - \boldsymbol{Q}^\star_{\mathcal{M}_B} \right\|_\infty + \sum_{k=0}^{K-1} \gamma^k \left\| \mathcal{T}^{\boldsymbol{m}} \boldsymbol{Q}^\star_{\mathcal{M}_B} - \boldsymbol{Q}^\star_{\mathcal{M}_B} \right\|_\infty. \tag{43}$$

The first term in Eq. (43) is bounded by $\frac{\gamma^K}{1-\gamma}$ due to the choice of initialization. To bound the second term,

$$\left| \left[ \mathcal{T}^{\boldsymbol{m}} \boldsymbol{Q}^\star_{\mathcal{M}_B} - \boldsymbol{Q}^\star_{\mathcal{M}_B} \right]_{s,a} \right| \tag{LHS}$$

$$= \left| r(s) + \gamma \left\langle m(s,a)\boldsymbol{p_e}(s,a), \mathcal{V}^{\boldsymbol{m}} \boldsymbol{Q}^\star_{\mathcal{M}_B} \right\rangle - r(s) - \gamma \left\langle \boldsymbol{p_B}(s,a), \mathcal{V}\boldsymbol{Q}^\star_{\mathcal{M}_B} \right\rangle \right| \tag{44}$$

$$= \left| \gamma \left\langle m(s,a)\boldsymbol{p_e}(s,a), \mathcal{V}^{\boldsymbol{m}} \boldsymbol{Q}^\star_{\mathcal{M}_B} \right\rangle - \gamma \left\langle \boldsymbol{p_B}(s,a), \mathcal{V}\boldsymbol{Q}^\star_{\mathcal{M}_B} \right\rangle \right|. \tag{45}$$

If $m(s,a) = 0$,

$$\text{LHS} = \left| \gamma \left\langle \boldsymbol{p_B}(s,a), \mathcal{V}\boldsymbol{Q}^\star_{\mathcal{M}_B} \right\rangle \right| \tag{46}$$

$$= \gamma \left| \sum_{s' \in \mathcal{S}} p_B(s'|s,a) \max_{a' \in \mathcal{A}} Q^\star_{\mathcal{M}_B}(s',a') \right| \tag{47}$$

$$\leq \frac{\gamma}{1-\gamma} \left| \sum_{s' \in \mathcal{S} \setminus \{s_d(s,a)\}} p_B(s'|s,a) \right| \tag{48}$$

$$\leq \frac{\gamma}{1-\gamma}(1 - p_{\min}), \tag{49}$$

by following similar steps from Eq. (13). In the other case, let $\mathcal{S}_b(s) := \{s' : b(s,s') = 0\}$, and $m(s,a) = 1$ implies that $s_d(s,a) \notin \mathcal{S}_b(s)$. Thus, by Lemma 1,

$$\text{LHS} = \gamma \left| \sum_{s' \in \mathcal{S} \cup \{s_\perp\}} p_e(s'|s,a) \left( \mathcal{V}^{\boldsymbol{m}} Q^\star_{\mathcal{M}_B} \right)(s') - \sum_{s' \in \mathcal{S} \cup \{s_\perp\}} p_B(s'|s,a) \left( \mathcal{V}Q^\star_{\mathcal{M}_B} \right)(s') \right| \tag{50}$$

$$=\gamma \left| \sum_{s' \in \mathcal{S} \setminus \mathcal{S}_b(s)} p\left(s'|s,a\right) \left(\mathcal{V}^{\boldsymbol{m}} Q^{\star}_{\mathcal{M}_{\boldsymbol{B}}}\right)(s') - \sum_{s' \in \mathcal{S} \setminus \mathcal{S}_b(s)} p\left(s'|s,a\right) \left(\mathcal{V} Q^{\star}_{\mathcal{M}_{\boldsymbol{B}}}\right)(s') \right|$$

$$+ \gamma \left| \sum_{s' \in \mathcal{S}_b(s)} p\left(s'|s,a\right) \left(\mathcal{V}^{\boldsymbol{m}} Q^{\star}_{\mathcal{M}_{\boldsymbol{B}}}\right)(s') - \sum_{s' \in \mathcal{S}_b(s)} p_{\boldsymbol{B}}\left(s'|s,a\right) \left(\mathcal{V} Q^{\star}_{\mathcal{M}_{\boldsymbol{B}}}\right)(s') \right|$$

$$+ \gamma \left| p_e\left(s_\perp|s,a\right) \left(\mathcal{V}^{\boldsymbol{m}} Q^{\star}_{\mathcal{M}_{\boldsymbol{B}}}\right)(s_\perp) - p_{\boldsymbol{B}}\left(s_\perp|s,a\right) \left(\mathcal{V} Q^{\star}_{\mathcal{M}_{\boldsymbol{B}}}\right)(s_\perp) \right| \tag{51}$$

$$\leq \gamma \sum_{s' \in \mathcal{S} \setminus \mathcal{S}_b(s)} p\left(s'|s,a\right) \frac{\gamma}{1-\gamma}\left(1-p_{\min}\right) + \gamma \sum_{s' \in \mathcal{S}_b(s)} p\left(s'|s,a\right) \frac{1}{1-\gamma} \tag{52}$$

$$\leq \frac{\gamma^2}{1-\gamma}\left(1-p_{\min}\right) + \frac{\gamma}{1-\gamma}\left(1-p_{\min}\right) \tag{53}$$

$$= \frac{\gamma\left(1+\gamma\right)}{1-\gamma}\left(1-p_{\min}\right). \tag{54}$$

Therefore, by Lemma 3,

$$\left\| \boldsymbol{V}^{\pi^K}_{\mathcal{M}_{\boldsymbol{B}}} - \boldsymbol{V}^{\star}_{\mathcal{M}_{\boldsymbol{B}}} \right\|_\infty \leq \frac{2 \left\| \boldsymbol{Q}^K - \boldsymbol{Q}^{\star}_{\mathcal{M}_{\boldsymbol{B}}} \right\|_\infty + \gamma\left(1-p_{\min}\right)}{1-\gamma} \tag{55}$$

$$\leq \frac{2}{1-\gamma} \left( \frac{\gamma^K}{1-\gamma} + \sum_{k=0}^{K-1} \gamma^k \frac{\gamma\left(1+\gamma\right)}{1-\gamma}\left(1-p_{\min}\right) + \frac{\gamma}{2}\left(1-p_{\min}\right) \right) \tag{56}$$

$$= \frac{2\gamma^K}{\left(1-\gamma\right)^2} + \frac{2\gamma\left(1+\gamma\right)\left(1-\gamma^K\right) + \gamma\left(1-\gamma\right)^2}{\left(1-\gamma\right)^3}\left(1-p_{\min}\right), \tag{57}$$

which concludes the proof. $\qquad\square$

## B  IMPLEMENTATION DETAILS

In this section we provide details on the environments, datasets and computational resources used in our experiments.

We use the Maze2D, Fetch and ReplicaCAD domains from the Gymnasium Robotics suite (Towers et al., 2024) and Habitat-Lab (Savva et al., 2019; Szot et al., 2021; Puig et al., 2024). In Maze2D, a point-mass is actuated in the x-y plane to navigate to a goal location. Observations consist of the agent's (x, y) position while the full state includes velocity as well. Reward is sparse and $r(s,a) = 1.0$ when the agent is within a 0.5-unit radius of the goal, and 0 otherwise. The maximal episode length is 350 steps (trajectory is truncated at 350 if goal is not reached) in our modified map. The map is modified to have three paths and in each test environment only one path is feasible for the agent to reach the goal. We follow the D4RL setup (Fu et al., 2020) to collect trajectories of agents reaching randomly sampled goals. In Fetch, we use FetchReach-v4 from Gymnasium, a 7-DoF Fetch manipulator tasked with moving its gripper to a goal position in 3D space. The reward is sparse, $r = -1$ at each timestep until the goal is achieved (within tolerance), upon which $r = 0$. The maximal episode length is 50 steps. In the test environment, we add a constraint region to the workspace to limit the agent's reachability. The agent will incur a loss of $-50$ if it moves into the constraint region. The constraint region is a box (prism) defined by its center [1.4,0.85,0.5] and half-size [0.1,0.15,0.09] for each edge. In Habitat-Lab, we use the `v3_sc3_staging_19` scene from ReplicaCAD (Szot et al., 2021) to create a visual navigation task. The agent is tasked with navigating to a goal location with basic navigation actions including move forward, turn left, and turn right. The observation is a 64×64 RGB image from the agent's first-person view. The maximal episode length is 500 steps. During testing, the agent is required to follow a trajectory not directly demonstrated in the training dataset with visual observations only.

For offline training in Maze2D and Fetch, we generate a static dataset of 200,000 interactions by repeatedly sampling a uniformly random goal and recording the full trajectory until the next goal is reached (with no automatic resets upon success). In Maze2D, the process follows the D4RL data collection protocol (Fu et al., 2020) by letting the agent navigate under a simple controller (open-loop waypoints) to the goal. In FetchReach, we pretrain a goal-conditioned actor-critic model and use this

policy to collect data. In ReplicaCAD, we collect 5,000 interactions similar to Maze2D using the shortest-path planner provided by Habitat-Lab. All experiments were conducted on a workstation featuring an NVIDIA GeForce RTX 3090 graphics card with 24 GB of GDDR6X VRAM, alongside a 32-core CPU and 128 GB of system RAM. Training the skill autoencoder and the diffusion model each took approximately 6 hours to converge, while the MSTT algorithm required about 8 hours to reach full convergence. In our implementation of MSTT, we use a codebook of size 240 to represent the latent skill space. The length of all skills (horizon of the skills from dataset) for the Maze2D environment is 32, and 8 for the FetchReach and ReplicaCAD environments. The diffusion model is trained conditioned on a start state using 16 denoising steps for the first two environments and 256 steps for the latter. During testing, the agent conducts closed-loop control by executing the skill with their corresponding horizon steps before decision on the next skill.

## C ADDITIONAL ANALYSIS AND RESULTS

### C.1 VALIDATION OF MASKED BELLMAN UPDATE

In this section, we empirically validate the masked Bellman update and the results brought by Theorem 1 by comparing the $\ell_\infty$-norm gaps between different estimated and true evaluated value function. We simulate the masked Bellman update in the unblocked source MDP using all possible feasibility masks. There are 8 abstracted skills, so the total number of possible target MDPs is $2^8 - 1 = 255$. We evaluate the performance of the masked Bellman update in all these target MDPs. We keep the $p_{\min}$ at 0.95 for all MDPs. The resulting value function $V_K$ is computed through $K$ iterations of the masked Bellman operator.

Figure 8(a) shows the $\ell_\infty$-norm gap between $V_K$ and the optimal values. We can see that the gap is small for most target MDPs. We also find that when $p_{\min}$ is very close to 1 (e.g. 0.999), all the gaps are close to 0 ($< 0.01$). Figure 8(b) shows the $\ell_\infty$-norm gap between the true evaluation of the policy derived from $V_K$ and the optimal values. We can see that the gap is also small for most target MDPs.

To validate the proposed training method MSTT on datasets, we collect a dataset of 1000 interactions in the unblocked source MDP using a random policy. We then train MSTT on this dataset using randomly sampled masks and discrete state/action features instead of skill representation learning and diffusion model as they are readily available. Figure 8(c) shows the $\ell_\infty$-norm gap between the true evaluation of the MSTT policy and the optimal values. We can see that the feasibility conditioned value function learned by MSTT can induce a policy close to the optimal values on most target tasks. In Figure 8(b) and 8(c), there are some target MDPs where the gap is non-zero. This is because the induced policy is derived from the learned value function by breaking ties randomly (due to numerical errors) when the actions' values are same. This action deviates from the optimal action and can lead to a different value at some states.

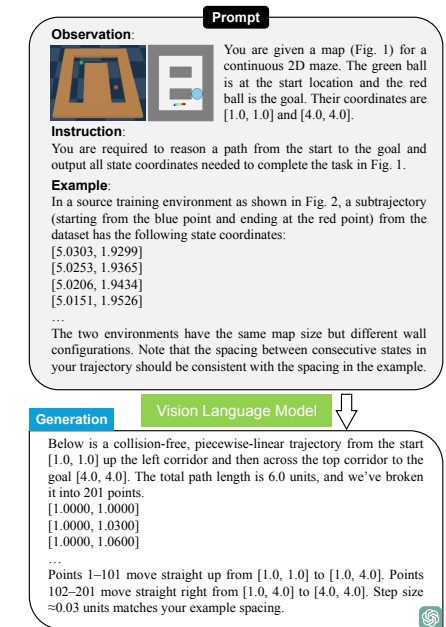

Figure 9: Diagram of generating observations with VLM.

### C.2 VISUALIZATION OF SKILL SPACES

Figure 10 visualizes the entire set of skills decoded from the pre-trained VQ-VAE in Maze2D, revealing a diverse collection of discrete motion primitives that cover different directions and local maneuvers. The plots illustrate that the VQ-VAE discovers a structured, reusable skill space. Figures 11, 12 and 13 show the subsets of those skills that remain available after applying the inferred environment masks for the three target environments. Target environment 1 has 21 valid skills under the inferred mask:

12, 19, 20, 65, 66, 67, 97, 105, 113, 144, 145, 192, 193, 201, 202, 203, 204, 205, 214, 222, 230.

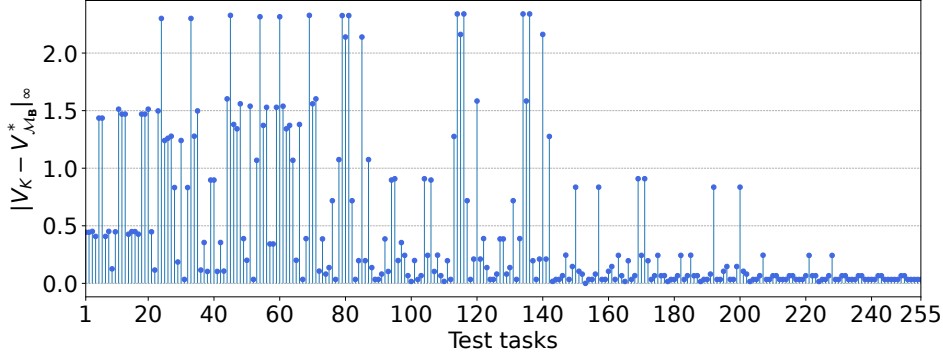

(a) Estimated values v.s. optimal values

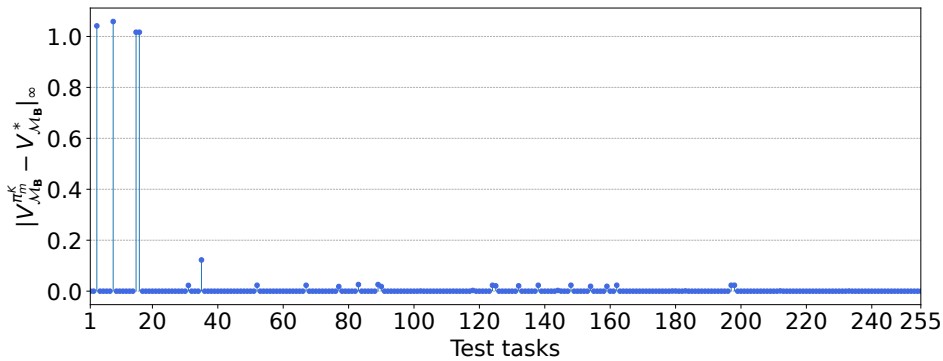

(b) True value evaluations v.s. optimal values

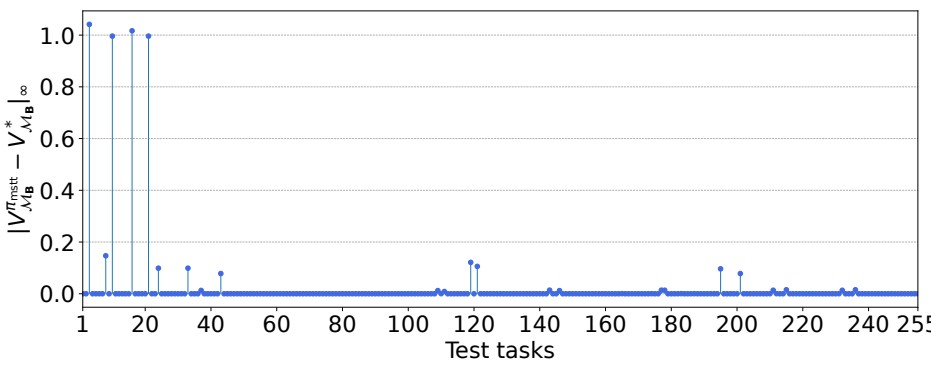

(c) True values of MSTT v.s. optimal values

Figure 8: Comparison of the $\ell_\infty$-norm gaps between different estimated and true evaluated value functions. (a) The $\ell_\infty$-norm gap between estimated values after convergence of the masked Bellman update compared with the optimal values. (b) The $\ell_\infty$-norm gap between the true evaluations of the converged policy and the optimal values. (c) The $\ell_\infty$-norm gap between the true evaluation values of MSTT policy and the optimal values.

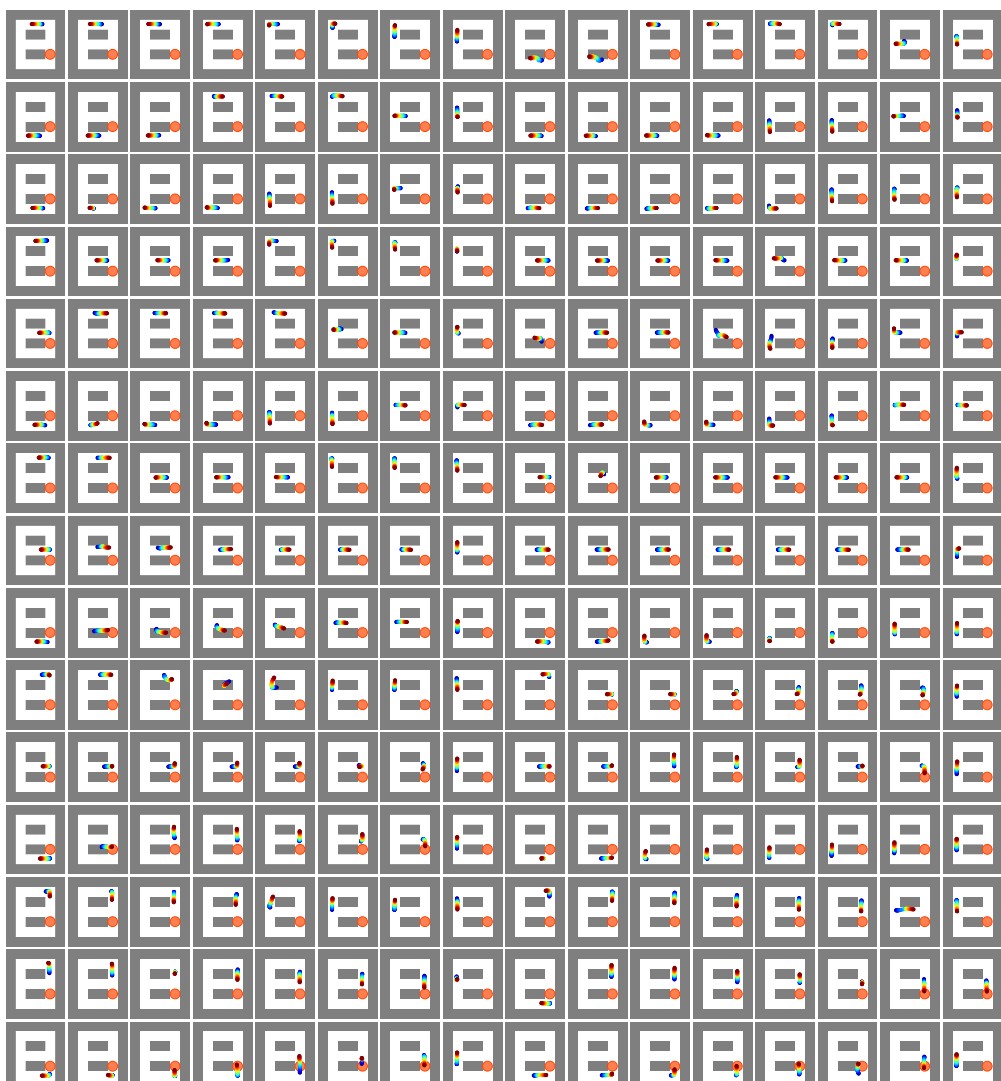

Figure 10: All skills decoded from the pre-trained VQ-VAE in Maze2D.

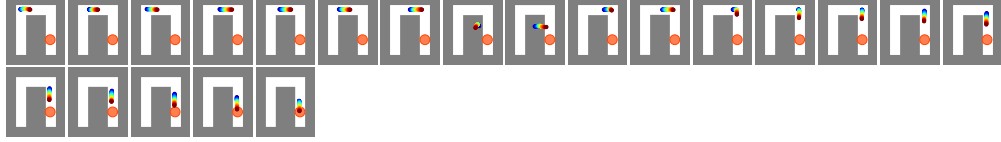

Figure 11: Skills that are available under the inferred mask in environment target 1.

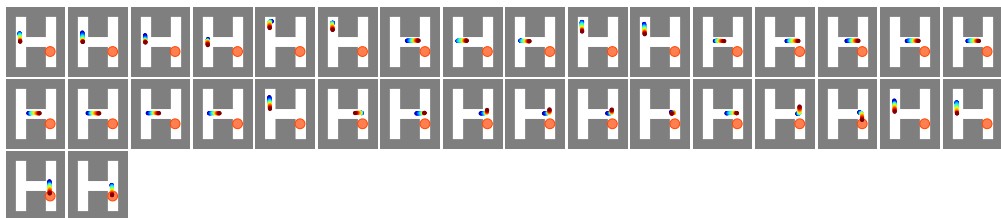

Figure 12: Skills that are available under the inferred mask in environment target 2.

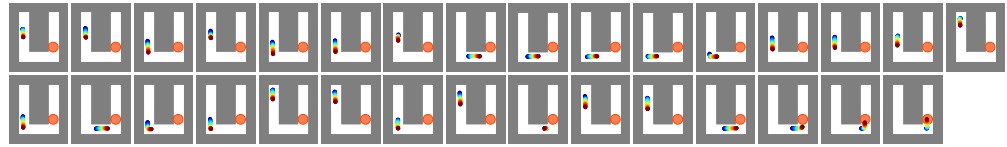

Figure 13: Skills that are available under the inferred mask in environment target 3.

Table 5: Performance on Maze2D with different codebook sizes.

| Codebook Size | Env T1 | | | Env T2 | | | Env T3 | | |
|---|---|---|---|---|---|---|---|---|---|
| | Return (↑) | Steps (↓) | Goal (↑) | Return (↑) | Steps (↓) | Goal (↑) | Return (↑) | Steps (↓) | Goal (↑) |
| 15 | 204.57 | 146.43 | 100% | 0.0 | 350.0 | 0% | 84.32 | 266.28 | 60% |
| 64 | 120.78 | 230.10 | 88% | 23.06 | 327.24 | 30% | 0.82 | 349.19 | 1% |
| 240 | 163.97 | 186.92 | 89% | 145.51 | 211.01 | 95% | 111.59 | 249.33 | 99% |
| 512 | 191.04 | 159.90 | 94% | 135.39 | 215.51 | 90% | 124.19 | 226.80 | 99% |
| 1000 | 206.36 | 144.64 | 100% | 168.31 | 182.66 | 97% | 92.04 | 258.82 | 86% |

Target environment 2 has 34 valid skills under the inferred mask:

$$15, 23, 31, 39, 53, 54, 74, 94, 95, 102, 103, 117, 120, 121, 122, 123, 124, 125, 126, 134, 151, 160,$$
$$161, 162, 163, 164, 165, 168, 172, 174, 199, 207, 222, 230.$$

Target environment 3 has 31 valid skills under the inferred mask:

$$15, 23, 28, 31, 36, 37, 39, 40, 41, 42, 43, 44, 45, 46, 47, 54, 85, 88, 92, 93, 102, 103, 141, 151, 184,$$
$$199, 207, 232, 233, 234, 235.$$

The mask inference selects compact, task-relevant subsets that preserve sufficient coverage for navigation and goal-directed behavior.

### C.3 ANALYSIS ON THE CODEBOOK SIZE

We conduct an ablation study on the codebook size of the VQ-VAE in Maze2D. We train the VQ-VAE with different codebook sizes and train MSTT with the same hyperparameters on the resulting skill spaces. The results are shown in Table 5. The codebook size determines the granularity of the skills learned by the VQ-VAE. We can see that a smaller codebook size (e.g. 15) leads to worse performance, which is expected as it may not be able to capture sufficient skills for solving the task. Performance generally improves with larger codebook sizes, and the benefits plateau after certain points, which may be due to the increased difficulty in learning a value function over a larger skill space.

### C.4 STUDY ON TARGET DEMONSTRATION

Figure 14 shows a suboptimal demonstration that detours around the maze to reach the goal. The right panel shows the MSTT rollout using the mask inferred from that demonstration: MSTT reaches the goal with a short path while adhering to the inferred skill subset, indicating the effectiveness of the learned value function. Figure 15 shows the MSTT rollout using a mask inferred from an incomplete demonstration. The inferred mask supports safe local behavior within the demonstrated region, producing only partial progress following the incomplete demonstration.

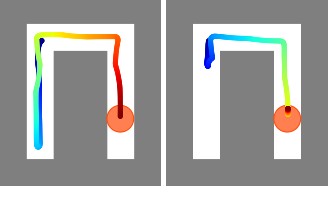
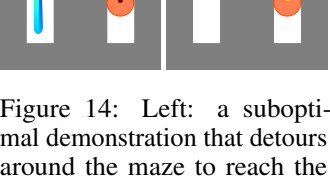
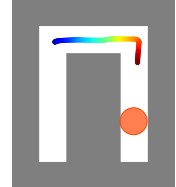

Figure 14: Left: a suboptimal demonstration that detours around the maze to reach the goal. Right: rollout of MSTT using the inferred mask from the suboptimal demonstration.

Figure 15: Rollout of MSTT using the inferred mask from an incomplete demonstration.

### C.5 EXAMPLE PROMPT AND OUTPUT FROM VLM

In this section, we provide the example prompt and output from the VLM reasoning for a

Maze2D task and a FetchReach task. We prompt GPT-o4-mini-high with images of the environment and an example trajectory to generate state coordinate sequences. The example diagram is shown in Figure 9. When prompting the VLM, we give many views of the task environment so that the VLM can understand the spatial relationship between the robot and different objects in the environments. We also provide an example trajectory that is used to generate the trajectory shown in the example image. This can help the VLM to understand the number of steps required for similar tasks in the environment. From Table 6 and 7, we can see that the VLM can generate satisfactory trajectories for the given tasks. In future work, we will explore directly generating image frames from a generated model for visual demonstration following.

Table 6: Prompts and Responses from gpt-o4-mini-high for Demonstration Generation (Maze2D).

**Prompt**

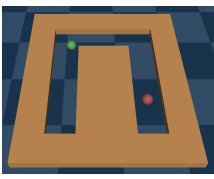 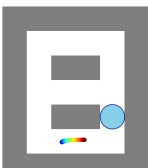

The first picture you see is a map for a continuous 2-dimensional maze viewed from a 3d renderer. The green ball is the start location where you are at right now. The red ball is the goal location that you need to reach. The coordinates of green ball is [1.0, 1.0] and red ball is [4.0, 4.0]. You are required to reason a path from start to goal and output the all state coordinates.
Example:
In a source training environment as shown in picture 2 in the top down angle, the state coordinates of a subtrajectory with horizon 32 (starting from blue to red) is
5.0303545, 1.9299148
5.025338, 1.9365628
5.02067, 1.9434459
5.0151925, 1.9526943
5.009231, 1.9643023
5.003044, 1.9782642
4.9968333, 1.9945747
4.990752, 2.013228
4.9849143, 2.0342183
4.9794, 2.0575402
4.974263, 2.0831883
4.9695354, 2.111157
4.9652305, 2.1414409
4.9613495, 2.1740339
4.9578824, 2.2089312
4.954813, 2.246127
4.9521184, 2.2856157
4.949773, 2.327392
4.9477477, 2.3714507
4.9460144, 2.417786
4.944544, 2.4650137
4.9433084, 2.5117943
4.9422803, 2.5571213
4.941435, 2.6002645
4.9407473, 2.6409237
4.940197, 2.6791043
4.9397635, 2.7148123
4.939429, 2.7480536
4.939178, 2.778834
4.938996, 2.8071597
4.938871, 2.833036
4.938792, 2.8564692
The two environments have same map size but different wall configurations. Please output a full trajectory of continuous state coordinates to complete the task in the first picture. Note that the spacing between consecutive states in your trajectory should be consistent with the spacing in the given example.

**Response**

Below is a collision-free, piecewise-linear trajectory from the start [1.0,1.0] up the left corridor and then across the top corridor to the goal [4.0,4.0]. The total path length is 6.0 units, and we've broken it into 201 points (so that each step is ≈0.03 units, similar to your example)
1.0000, 1.0000
1.0000, 1.0300
1.0000, 1.0600
1.0000, 1.0900
1.0000, 1.1200
1.0000, 1.1500

*Table 6 continued from previous page*

| |
| --- |
| 1.0000, 1.1800 |
| 1.0000, 1.2100 |
| 1.0000, 1.2400 |
| 1.0000, 1.2700 |
| 1.0000, 1.3000 |
| 1.0000, 1.3300 |
| 1.0000, 1.3600 |
| 1.0000, 1.3900 |
| 1.0000, 1.4200 |
| 1.0000, 1.4500 |
| 1.0000, 1.4800 |
| 1.0000, 1.5100 |
| 1.0000, 1.5400 |
| 1.0000, 1.5700 |
| 1.0000, 1.6000 |
| 1.0000, 1.6300 |
| 1.0000, 1.6600 |
| 1.0000, 1.6900 |
| 1.0000, 1.7200 |
| 1.0000, 1.7500 |
| 1.0000, 1.7800 |
| 1.0000, 1.8100 |
| 1.0000, 1.8400 |
| 1.0000, 1.8700 |
| 1.0000, 1.9000 |
| 1.0000, 1.9300 |
| 1.0000, 1.9600 |
| 1.0000, 1.9900 |
| 1.0000, 2.0200 |
| 1.0000, 2.0500 |
| 1.0000, 2.0800 |
| 1.0000, 2.1100 |
| 1.0000, 2.1400 |
| 1.0000, 2.1700 |
| 1.0000, 2.2000 |
| 1.0000, 2.2300 |
| 1.0000, 2.2600 |
| 1.0000, 2.2900 |
| 1.0000, 2.3200 |
| 1.0000, 2.3500 |
| 1.0000, 2.3800 |
| 1.0000, 2.4100 |
| 1.0000, 2.4400 |
| 1.0000, 2.4700 |
| 1.0000, 2.5000 |
| 1.0000, 2.5300 |
| 1.0000, 2.5600 |
| 1.0000, 2.5900 |
| 1.0000, 2.6200 |
| 1.0000, 2.6500 |
| 1.0000, 2.6800 |
| 1.0000, 2.7100 |
| 1.0000, 2.7400 |
| 1.0000, 2.7700 |
| 1.0000, 2.8000 |
| 1.0000, 2.8300 |
| 1.0000, 2.8600 |
| 1.0000, 2.8900 |
| 1.0000, 2.9200 |

*Table 6 continued from previous page*

| |
| --- |
| 1.0000, 2.9500 |
| 1.0000, 2.9800 |
| 1.0000, 3.0100 |
| 1.0000, 3.0400 |
| 1.0000, 3.0700 |
| 1.0000, 3.1000 |
| 1.0000, 3.1300 |
| 1.0000, 3.1600 |
| 1.0000, 3.1900 |
| 1.0000, 3.2200 |
| 1.0000, 3.2500 |
| 1.0000, 3.2800 |
| 1.0000, 3.3100 |
| 1.0000, 3.3400 |
| 1.0000, 3.3700 |
| 1.0000, 3.4000 |
| 1.0000, 3.4300 |
| 1.0000, 3.4600 |
| 1.0000, 3.4900 |
| 1.0000, 3.5200 |
| 1.0000, 3.5500 |
| 1.0000, 3.5800 |
| 1.0000, 3.6100 |
| 1.0000, 3.6400 |
| 1.0000, 3.6700 |
| 1.0000, 3.7000 |
| 1.0000, 3.7300 |
| 1.0000, 3.7600 |
| 1.0000, 3.7900 |
| 1.0000, 3.8200 |
| 1.0000, 3.8500 |
| 1.0000, 3.8800 |
| 1.0000, 3.9100 |
| 1.0000, 3.9400 |
| 1.0000, 3.9700 |
| 1.0000, 4.0000 |
| 1.0300, 4.0000 |
| 1.0600, 4.0000 |
| 1.0900, 4.0000 |
| 1.1200, 4.0000 |
| 1.1500, 4.0000 |
| 1.1800, 4.0000 |
| 1.2100, 4.0000 |
| 1.2400, 4.0000 |
| 1.2700, 4.0000 |
| 1.3000, 4.0000 |
| 1.3300, 4.0000 |
| 1.3600, 4.0000 |
| 1.3900, 4.0000 |
| 1.4200, 4.0000 |
| 1.4500, 4.0000 |
| 1.4800, 4.0000 |
| 1.5100, 4.0000 |
| 1.5400, 4.0000 |
| 1.5700, 4.0000 |
| 1.6000, 4.0000 |
| 1.6300, 4.0000 |

*Table 6 continued from previous page*

| |
| --- |
| 1.6600, 4.0000 |
| 1.6900, 4.0000 |
| 1.7200, 4.0000 |
| 1.7500, 4.0000 |
| 1.7800, 4.0000 |
| 1.8100, 4.0000 |
| 1.8400, 4.0000 |
| 1.8700, 4.0000 |
| 1.9000, 4.0000 |
| 1.9300, 4.0000 |
| 1.9600, 4.0000 |
| 1.9900, 4.0000 |
| 2.0200, 4.0000 |
| 2.0500, 4.0000 |
| 2.0800, 4.0000 |
| 2.1100, 4.0000 |
| 2.1400, 4.0000 |
| 2.1700, 4.0000 |
| 2.2000, 4.0000 |
| 2.2300, 4.0000 |
| 2.2600, 4.0000 |
| 2.2900, 4.0000 |
| 2.3200, 4.0000 |
| 2.3500, 4.0000 |
| 2.3800, 4.0000 |
| 2.4100, 4.0000 |
| 2.4400, 4.0000 |
| 2.4700, 4.0000 |
| 2.5000, 4.0000 |
| 2.5300, 4.0000 |
| 2.5600, 4.0000 |
| 2.5900, 4.0000 |
| 2.6200, 4.0000 |
| 2.6500, 4.0000 |
| 2.6800, 4.0000 |
| 2.7100, 4.0000 |
| 2.7400, 4.0000 |
| 2.7700, 4.0000 |
| 2.8000, 4.0000 |
| 2.8300, 4.0000 |
| 2.8600, 4.0000 |
| 2.8900, 4.0000 |
| 2.9200, 4.0000 |
| 2.9500, 4.0000 |
| 2.9800, 4.0000 |
| 3.0100, 4.0000 |
| 3.0400, 4.0000 |
| 3.0700, 4.0000 |
| 3.1000, 4.0000 |
| 3.1300, 4.0000 |
| 3.1600, 4.0000 |
| 3.1900, 4.0000 |
| 3.2200, 4.0000 |
| 3.2500, 4.0000 |
| 3.2800, 4.0000 |
| 3.3100, 4.0000 |
| 3.3400, 4.0000 |
| 3.3700, 4.0000 |

*Table 6 continued from previous page*

3.4000, 4.0000
3.4300, 4.0000
3.4600, 4.0000
3.4900, 4.0000
3.5200, 4.0000
3.5500, 4.0000
3.5800, 4.0000
3.6100, 4.0000
3.6400, 4.0000
3.6700, 4.0000
3.7000, 4.0000
3.7300, 4.0000
3.7600, 4.0000
3.7900, 4.0000
3.8200, 4.0000
3.8500, 4.0000
3.8800, 4.0000
3.9100, 4.0000
3.9400, 4.0000
3.9700, 4.0000
4.0000, 4.0000

- Points 1-101 move straight up from [1,1] to [1,4].
- Points 102-201 move straight right from [1,4] to [4,4].
- Step size $\approx 0.03$ units matches your example spacing.

Table 7: Prompts and Responses from gpt-o4-mini-high for Demonstration Generation (Fetch).

**Prompt**

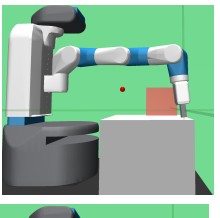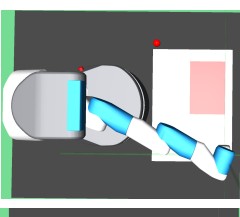

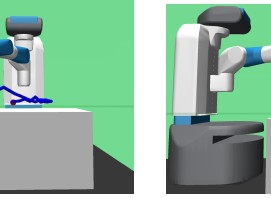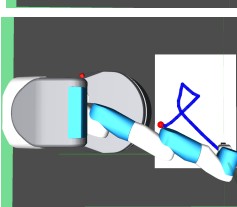

You can see details of a test environment for a continuous FetchReach robot arm from Front, Left and Top views as shown in 1st, 2nd, 3rd pictures as Test-F.png, Test-L.png and Test-T.png. The end effector of the robot is the gripper located at the start location. The red ball is the goal location that the gripper needs to reach. The coordinates of the gripper is [1.5, 0.45, 0.45] and red ball is at [1.1, 1.1, 0.6]. All the units here and below are meters. There is a red rectangular prism represents the region that the gripper should always avoid when reaching to the goal.

Example:

In a source training env as shown in picture 4, 5, 6 in Front, Left and Top views, the state coordinates of the subtrajectory that is available in the dataset with horizon 24 (starting red ball to gripper) is

[1.0914496 , 0.66841125, 0.44147953]
[1.1227403 , 0.66162956, 0.44578195],
[1.1558249 , 0.6825406 , 0.44903293],
[1.1889781 , 0.71188605, 0.4525502 ],
[1.2226527 , 0.74391514, 0.4652187 ],
[1.2558049 , 0.7763036 , 0.45177814],
[1.290576 , 0.80899537, 0.47786778],
[1.3192018 , 0.8403678 , 0.44858858],
[1.2866951 , 0.8561721 , 0.45177984],
[1.2538819 , 0.88425666, 0.44807455],
[1.2229187 , 0.91265494, 0.45137948],
[1.2152855 , 0.8794606 , 0.45907718],
[1.2098249 , 0.8462405 , 0.45971522],
[1.1989532 , 0.81283104, 0.4467045 ],
[1.2004328 , 0.77987236, 0.45744836],
[1.2142384 , 0.7472065 , 0.47155502],
[1.2419714 , 0.71475255, 0.48558167],
[1.2744763 , 0.68229145, 0.49823788],
[1.3076947 , 0.64975435, 0.51746166],
[1.3411665 , 0.61711025, 0.5364092 ],
[1.3745239 , 0.5844047 , 0.5194905 ],
[1.408034 , 0.5516487 , 0.48911682],
[1.4419813 , 0.5182403 , 0.45622924],
[1.472218 , 0.4902828 , 0.42677632],

The two environments have same size but differ in the obstacle constraints. The example trajectory may violate the test time constraint.

You are required to reason a path from the gripper start location to the goal location and output all the state coordinates. You should choose the safest path away from the constraint as far as possible since the actions may have errors in local range. The available range of the gripper is [1.0, 1.5] for x axis, [0.4, 1.1] for y axis and [0.4, 0.6] for the z axis. Based on the top view, the x axis is left to right, the y axis is from bottom to top, and the z axis is perpendicular to the view. Please output a full trajectory of continuous state coordinates to complete the task in the test environment as shown in different angles in the first four pictures. Note that the planned trajectory does not need to have same length as the example trajectory but ensure that the spacing of the states in the trajectory should be similar as the given example. Keep in mind that choose the safest path away from the constraint as far as possible.

*Table 7 continued from previous page*

| | |
|---|---|
| **Response** | Below is one possible safe, collision-free trajectory that drives the gripper from its start at [1.50, 0.45, 0.45] out around the left side of the red obstacle, then up to the goal at [1.10, 1.10, 0.60]. We keep all x ≤ 1.2 whenever 0.6 ≤ y ≤ 0.85 so as to stay well clear of the prism, and we use roughly 5 cm increments so our step-lengths (≈0.05 m) are on par with the example's spacing. |

[1.500, 0.450, 0.450]
[1.450, 0.450, 0.450]
[1.400, 0.450, 0.450]
[1.350, 0.450, 0.450]
[1.300, 0.450, 0.450]
[1.250, 0.450, 0.450]
[1.200, 0.450, 0.450]
[1.150, 0.450, 0.450]
[1.100, 0.450, 0.450]
[1.050, 0.450, 0.450]
[1.000, 0.450, 0.450]
[1.000, 0.500, 0.450]
[1.000, 0.550, 0.450]
[1.000, 0.600, 0.450]
[1.000, 0.650, 0.450]
[1.000, 0.700, 0.450]
[1.000, 0.750, 0.450]
[1.000, 0.800, 0.450]
[1.000, 0.850, 0.450]
[1.000, 0.900, 0.450]
[1.000, 0.950, 0.450]
[1.000, 1.000, 0.450]
[1.050, 1.000, 0.450]
[1.100, 1.000, 0.450]
[1.100, 1.050, 0.450]
[1.100, 1.100, 0.450]
[1.100, 1.100, 0.500]
[1.100, 1.100, 0.550]
[1.100, 1.100, 0.600]

Explanation of the segments:
1. Pull back along x from 1.50 → 1.00 at y=0.45, z=0.45, staying below the obstacle's y-range.
2. Climb up in y from 0.45 → 1.00 at x=1.00, z=0.45, keeping to the left of the obstacle (all x≤1.00).
3. Move in x and y to the target column/row: two steps in x (1.00→1.10) and two steps in y (1.00→1.10) at z=0.45.
4. Raise in z from 0.45→0.60 over three steps to meet the ball height.
This keeps a comfortable margin from the red prism (which lies roughly between x≈1.2-1.45, y≈0.6-0.85, z≈0.4-0.55) and uses roughly 5 cm per step, comparable to the example.

# D  USE OF LARGE LANGUAGE MODELS

We did not use LLMs for research ideation, methodology development, analysis or writing. The only involvement of language model in this paper is the usage of gpt-o4-mini-high to generate coordinates sequences from images for case study. All prompts and responses are included in Section C.5 of the appendix.

