# OpenReview forum: "Masked Skill Token Training for Hierarchical Off-Dynamics Transfer"
_ICLR.cc/2026/Conference — ICLR 2026 Poster_

### Official Review · Reviewer_VHqp · 2025-10-27

**Soundness:** 3
**Presentation:** 3
**Contribution:** 3
**Rating:** 6
**Confidence:** 4

**Summary:**

This paper introduces MSTT, a hierarchical RL framework for off-dynamics policy transfer. The agent is trained entirely offline on data from a source environment and must adapt to a target environment with structural changes (e.g., blocked paths) using only a single, observation-only demonstration. MSTT first learns a discrete skill vocabulary by tokenizing trajectory segments with a VQ-VAE. It then trains a skill-conditioned Q-function using a novel masked Bellman operator, which simulates dynamics shifts during training by randomly making subsets of skills unavailable. At test time, a feasibility mask is inferred from the skills observed in the demonstration, and the agent plans using the learned Q-function conditioned on this mask. A diffusion model generates trajectories corresponding to feasible skills. The method is shown to significantly outperform baselines on tasks with altered dynamics in Maze2D, FetchReach, and Habitat.

**Strengths:**

- Originality: The primary strength is the novel formulation of the masked Bellman operator for learning a feasibility-conditioned value function. This provides an elegant way to simulate a wide range of dynamics shifts during offline training without requiring multi-environment data.

- Significance: The paper tackles the critical problem of generalization in RL under a practical, low-supervision setting. The ability to adapt from a single, action-free demonstration makes the approach applicable to real-world scenarios where target environment interaction is costly or unsafe.

- Clarity: The paper's exposition is good. Complex ideas are broken down into understandable components, and the writing is concise.

- Empirical Quality: The experiments are thorough and convincingly demonstrate the superiority of MSTT over strong baselines in diverse and challenging environments, including continuous control and visual navigation.

**Weaknesses:**

- Mask Inference: The test-time mask is inferred by assuming any skill observed in the single demonstration is feasible, and all others are not. This is a strong heuristic that may be brittle. If the demonstration is suboptimal or does not exhaustively cover all feasible and necessary skills for a given task, the agent's capability will be unnecessarily constrained.

- Scalability of Mask Sampling: The training relies on random sampling of skill masks from a space of size 2^L, where L is the codebook size. For the reported L=240, this space is astronomically large. While the paper shows this works, there is no analysis on why random sampling is sufficient. It is plausible the method's effectiveness could degrade for problems requiring larger or less structured skill vocabularies where random sampling provides poor coverage.

- Scope of Dynamics Shifts: The method is tailored for structural dynamics changes that render skills binary (feasible/infeasible). It is less clear how it would adapt to continuous dynamics shifts (e.g., changes in mass or friction), which is a limitation on the "off-dynamics" claim. The authors acknowledge this in a footnote.

**Questions:**

- Robustness of Mask Inference: How sensitive is the method to the quality of the demonstration? For example, in Maze2D, what if the demonstration covers only the first half of the only feasible path? Can the agent still reach the goal, or does the incomplete mask render the problem unsolvable?
- On the Sufficiency of Random Masking: Could you provide more intuition as to why random mask sampling during training is effective, given the combinatorial explosion of the mask space? Does the value function learn to generalize across masks with similar properties, or does the structure of the skill space make many masks irrelevant?
- Beyond Binary Feasibility: The current model assumes skills are either feasible or not. Have you considered extending this to handle dynamics shifts that make skills more costly or stochastic, rather than impossible? For example, by learning a mask that represents a cost multiplier instead of a binary indicator.
- Influence of VQ-VAE: The performance seems highly dependent on the quality of the learned skill codebook. Could you comment on how performance changes with the codebook size and provide any qualitative analysis of the learned skills to show they are semantically meaningful?

---

> ### Author Response · Authors · 2025-11-22
> **Official Comment by Authors (1/3)**
>
> Thank you very much for your insightful and constructive feedback on our submission. The following are our detailed responses.
>
> ### Weakness 1:
>
> > "Mask Inference: The test-time mask is inferred by assuming any skill observed in the single demonstration is feasible, and all others are not. This is a strong heuristic that may be brittle. If the demonstration is suboptimal or does not exhaustively cover all feasible and necessary skills for a given task, the agent's capability will be unnecessarily constrained."
>
> ### Question 1:
>
> > "Robustness of Mask Inference: How sensitive is the method to the quality of the demonstration? For example, in Maze2D, what if the demonstration covers only the first half of the only feasible path? Can the agent still reach the goal, or does the incomplete mask render the problem unsolvable?"
>
> **Response**:
>
> We agree with the reviewer that MSTT indeed assumes that (1) target demonstration can accomplish the task, and (2) any skill observed in the single demonstration is feasible and available in the source dataset. We would like to claim it is a feature of MSTT that it values the **safety requirement induced by the demonstration**.
>
> 1. Incomplete skill: MSTT considers any skill that is not present in the demonstration to be unsafe to use, as using it may cause damage. Therefore, MSTT is designed to safely adapt to the target domain. For incomplete demonstration, MSTT with constrained sampling will not achieve the goal. For example as shown in `Figure 16` in the appendix (`Section D.3`), if the demonstration only covers the first half of the feasible path, MSTT will only leverage available skills. But if we ignore the sampling constraints induced by the mask, the agent will likely finish the task with the prior knowledge learned from source task. However, this may be not safe, for example, in fetch task the agent may enter the dangerous obstacle area.
>
> 2. Infeasible skill: if the demonstration has new skills that are not present in the source dataset, then inferring the mask from it will lead to random behavior. MSTT works in an offline manner, and it does not have online exploration. If a user wants the agent to achieve a task in the target domain that requires new skills, additional data collection and skill learning would be necessary. Enabling continual learning on new skill trajectory could potentially help discover the optimal solution in such cases, which is an interesting direction, and we would like to explore as future work.
>
> ---

---

> > ### Author Response · Authors · 2025-11-22
> > **Official Comment by Authors (2/3)**
> >
> > ### Weakness 2:
> >
> > > "Scalability of Mask Sampling: While the paper shows this works, there is no analysis on why random sampling is sufficient."
> >
> > ### Question 2:
> >
> > > "On the Sufficiency of Random Masking: Could you provide more intuition as to why random mask sampling during training is effective"
> >
> > **Response**:
> >
> > We thank the reviewer for this insightful comment. We find that skills are well-structured and random mask sampling exposes the critic to a diverse set of masking patterns.
> >
> > 1. We find that the skills learned by VQ-VAE are well-structured and the transitions between skills are sparse. `Figure 13` in the appendix (`Section D.2`) illustrates the transition matrix of the skill tokens extracted from the source dataset. The clear banded structure indicates that only a small subset of skills is likely to transition to each other. This property helps mitigate the risk of overfitting to partial masking patterns, as many masking patterns become irrelevant due to the sparse transitions.<br>
> > We evaluate the generalization over masks by validating on a fixed set of masks that are not used for sampling during training. `Section D.2` in the appendix presents the results shown in `Figure 14`, where the performance on the validation masks set is comparable to that on the training masks, indicating good generalization.
> >
> > 2. Instead of using random masking, we have conducted experiments where the masking process is dataset dependent by following similar distributions of skills as the source dataset. We label all the skills (short trajectories) in the source dataset with their corresponding skill token indices, which leads to sequences of skill indices and an empirical transition matrix estimated by counting the occurrences of skill transitions. A mask can be generated by labeling the available skills sampled from the transition matrix with certain steps. As shown in **Table 1** below, MSTT with learned masking achieves reasonable performance but is still outperformed by random masking. We hypothesize that this is because the random masking strategy provides sufficient coverage of the skill space during training to regularize the critic.
> >
> > | Table 1 (Reviewer-VHqp) | | Env T1 | | | Env T2 | | | Env T3 | |
> > |---|:---:|:---:|:---:|:---:|:---:|:---:|:---:|:---:|:---:|
> > | | Return ($\uparrow$) | Steps ($\downarrow$) | Goal ($\uparrow$) | Return ($\uparrow$) | Steps ($\downarrow$) | Goal ($\uparrow$) | Return ($\uparrow$) | Steps ($\downarrow$) | Goal ($\uparrow$) |
> > | Random Masking | 163.97 | 186.92 | 89% | 145.51 | 211.01 | 95% | 111.59 | 249.33 | 99% |
> > | Learned Masking | 154.68 | 196.10 | 78% | 134.56 | 216.28 | 84% | 82.93 | 267.91 | 84% |
> >
> > These results and our prior experiments demonstrate that random mask sampling provides a simple yet effective way to expose the critic to a diverse set of masking patterns. For larger number of skills, one can learn an autoregressive model over the skill indices on the datasets and sample masks from the model. We view large-scale codebook learning and mask sampling strategy as an interesting direction and believe that our work provides insightful findings for this direction.
> >
> > ---
> >
> > ### Weakness 3:
> >
> > > "Scope of Dynamics Shifts: The method is tailored for structural dynamics changes that render skills binary (feasible/infeasible)."
> >
> > **Response**:
> >
> > We thank the reviewer for this insightful comment. MSTT is primarily designed to handle structural dynamics changes that lead to binary feasibility of skills. However, we believe that MSTT can be extended to address certain types of continuous dynamics shifts. To this end, we create a hybrid dataset that combines both structural and continuous dynamics changes, where the trajectories are collected with the mass of the agent being 0.1x or 10x of the original mass.
> >
> > | Table 2 (Reviewer-VHqp) | | Env T1 | | | Env T2 | | | Env T3 | |
> > |---|:---:|:---:|:---:|:---:|:---:|:---:|:---:|:---:|:---:|
> > | | Return ($\uparrow$) | Steps ($\downarrow$) | Goal ($\uparrow$) | Return ($\uparrow$) | Steps ($\downarrow$) | Goal ($\uparrow$) | Return ($\uparrow$) | Steps ($\downarrow$) | Goal ($\uparrow$) |
> > | Hybrid Dataset | 159.83 | 191.15 | 88% | 142.94 | 208.04 | 88% | 84.45 | 266.45 | 90% |
> >
> > **Table 2** summarizes the results, showing that MSTT can still effectively adapt to the dynamics shifts by inferring the skills from the target demonstration. We would like to explore this direction further in future work by extending the dynamics shift to more fine-grained changes with larger codebook sizes.
> >
> > ---

---

> > > ### Author Response · Authors · 2025-11-22
> > > **Official Comment by Authors (3/3)**
> > >
> > > ### Question 3:
> > >
> > > > "Beyond Binary Feasibility: The current model assumes skills are either feasible or not. Have you considered extending this to handle dynamics shifts that make skills more costly or stochastic, rather than impossible? For example, by learning a mask that represents a cost multiplier instead of a binary indicator."
> > >
> > > **Response**:
> > >
> > > We appreciate the reviewer's suggestion. Extending MSTT to handle dynamics shifts that introduce cost multipliers or stochasticity is indeed an interesting direction. One potential approach could involve updating the mask transition matrix in a Bayesian manner and use the posterior after observing the demonstration, allowing the model to capture uncertainty in skill feasibility. We will explore this as future work.
> > >
> > > ---
> > >
> > > ### Question 4:
> > >
> > > > "Influence of VQ-VAE: The performance seems highly dependent on the quality of the learned skill codebook. Could you comment on how performance changes with the codebook size and provide any qualitative analysis of the learned skills to show they are semantically meaningful?"
> > >
> > > **Response**:
> > >
> > > The codebook size determines the granularity of the skills learned by the VQ-VAE. A smaller codebook size leads to coarser skills that are less expressive, while a larger codebook size allows for a more diverse set of skills, which can potentially improve performance by providing finer control over the agent's behavior. However, it also increases the complexity of the skill space, which may make it more challenging for the critic to learn effective value estimates.
> > >
> > > We have conducted experiments with different codebook sizes (e.g., 15, 64, 240, 512 and 1000) and **Table 3** below summarizes the results on Maze2D environment. We observe that performance generally improves with larger codebook sizes, and the benefits plateau after certain points.
> > >
> > > | Table 3 (Reviewer-VHqp) | | Env T1 | | | Env T2 | | | Env T3 | |
> > > |---|:---:|:---:|:---:|:---:|:---:|:---:|:---:|:---:|:---:|
> > > | Codebook Size | Return ($\uparrow$) | Steps ($\downarrow$) | Goal ($\uparrow$) | Return ($\uparrow$) | Steps ($\downarrow$) | Goal ($\uparrow$) | Return ($\uparrow$) | Steps ($\downarrow$) | Goal ($\uparrow$) |
> > > | 15 | 204.57 | 146.43 | 100% | 0.0 | 350.0 | 0% | 84.32 | 266.28 | 60% |
> > > | 64 | 120.78 | 230.1 | 88% | 23.06 | 327.24 | 30% | 0.82 | 349.19 | 1% |
> > > | 240 | 163.97 | 186.92 | 89% | 145.51 | 211.01 | 95% | 111.59 | 249.33 | 99% |
> > > | 512 | 191.04 | 159.9 | 94% | 135.39 | 215.51 | 90% | 124.19 | 226.8 | 99% |
> > > | 1000 | 206.36 | 144.64 | 100% | 168.31 | 182.66 | 97% | 92.04 | 258.82 | 86% |
> > >
> > > We have added visualizations of all the skills and unmasked skills in different target domains in the revised manuscript (`Section D.1`). `Figure 9` illustrates the skills decoded from the pre-trained VQ-VAE model, demonstrating its semantic meanings. `Figure 10, 11, 12` shows the skills available under the inferred mask from the target demonstration. They correspond well with the environmental changes. These visualizations provide qualitative and interpretable evidence that MSTT effectively identifies and utilizes feasible skills in the target domains.
> > >
> > > ---
> > >
> > > We sincerely thank you for your encouraging evaluation and thoughtful comments. Your recognition of our work’s contributions is highly motivating, and your suggestions have helped us further strengthen the clarity and rigor of the manuscript.

---

### Official Review · Reviewer_DAdX · 2025-10-29

**Soundness:** 3
**Presentation:** 3
**Contribution:** 3
**Rating:** 6
**Confidence:** 4

**Summary:**

This paper introduces Masked Skill Token Training (MSTT), a hierarchical reinforcement learning framework for off-dynamics policy transfer in a fully offline setting. The core idea is to adapt an agent trained in a source environment to a target environment with new constraints (like a blocked passage) using only a single, observation-only demonstration for adaptation.

MSTT works in three phases:
1.  **Skill Tokenization:** It uses a VQ-VAE to learn a discrete codebook of "skills" from fixed-length, unsupervised trajectory segments.
2.  **Masked Critic Training:** It trains a "feasibility-conditioned" $Q$-function. This is the clever part: it uses a novel masked Bellman operator and randomly samples binary skill masks during training to simulate a wide variety of potential dynamics shifts.
3.  **Adaptation & Execution:** At test time, it infers a single, fixed feasibility mask by tokenizing the provided demonstration. A diffusion-based generative policy then proposes candidate trajectories, which are filtered by this mask before being executed.

The authors provide a theoretical analysis for their masked Bellman operator and demonstrate strong empirical performance on several control and navigation tasks.

**Strengths:**

1.  **Novel and Practical Problem:** The paper tackles a well-motivated and important problem: adapting offline-trained agents to new environmental constraints without costly interaction or action labels.
2.  **Elegant Core Idea:** The "masked Bellman operator" is the paper's strongest asset. The insight to train a single critic conditioned on a binary feasibility mask $m$, and to generalize this critic by randomly sampling $m$ during training, is a new idea.
3.  **Sound Theoretical Core:** The method is justified by a novel theoretical analysis. The core contribution is Lemma 1 (we have checked the correctness), which bounds the approximation error of the masking approach and links it directly to skill determinism ($p_{min} \to 1$). While the associated proofs for Lemma 2 and Lemma 3 are based on standard Q-learning convergence analyses, they are essential for the completeness of the main theorem, which is both novel and well-motivated.
4.  **Dominant Empirical Success:** The framework is not just a theory. It achieves near-expert performance and massively outperforms strong baselines across all tasks. The FetchReach result, where MSTT is the *only* method to both succeed and avoid the new constraint (0% failure rate), is particularly striking.

**Weaknesses:**

1.  **Misleading Presentation of "Options":** The paper's framing around the "options" framework is a bit oversell. The method's theory and implementation do not address true temporal abstraction (i.e., SMDPs) but are instead restricted to a fixed-horizon, augmented action-space MDP. While the proof for the masked MDP's bounded convergence is novel, this limitation of the theoretical contribution to a standard MDP, rather than a more general SMDP, constrains the work's theoretical novelty. This critical simplifying assumption—the fixed skill horizon—might be better stated clearly upfront in the main text, not relegated to a footnote or appendix. We would encourage the authors to adopt the more rigorous and accurate term "augmented action-space MDP" throughout the paper; doing so would significantly improve the clarity and precision of the contribution.

2.  **Simplistic Mask Inference:** The test-time mask inference is a simple "set $m(z)=1$ if $z$ is in the demo." This assumes the single demonstration is both correct and comprehensive. The paper seems doesn't explore how robust this is to suboptimal or noisy demonstrations. And a soft version seems to be more plausible than hard assignment for real world application.

3.  **Potential Inefficiency:** The pipeline of a VQ-VAE encoder, a diffusion model policy and the complexity of sampling mechanism seems limited to scale to large scale applications. The test-time adaptation (Algorithm 2) is a rejection sampler. In highly constrained target environments (where most $m(z)=0$), this could be extremely inefficient, requiring many samples from the diffusion model to find one valid option. So the scalability feels a bit limited.

4. A closely related work with potential theoretical improvement is if the masked value theorem applied to HiT-MDP then the claim of options will hold. Also skill discovery algo will be simplified and generalized. Authors may refer to

Li, Chang, Dongjin Song, and Dacheng Tao. "The skill-action architecture: Learning abstract action embeddings for reinforcement learning." (2021).

Li, Chang, Dongjin Song, and Dacheng Tao. "Hit-MDP: learning the SMDP option framework on MDPs with hidden temporal embeddings." The Eleventh International Conference on Learning Representations. 2023.

for an efficient and scalable option embedding discovery framework. An offline algo vmoc is derived here:

Li, Chang, et al. "Learning Temporal Abstractions via Variational Homomorphisms in Option-Induced Abstract MDPs." arXiv preprint arXiv:2507.16473 (2025).

**Questions:**

1.  Given that the theory and implementation rely on fixed-horizon skills $H$, would the authors be willing to clarify this in the main text (e.g., by framing it as an "augmented action-space MDP") to avoid confusion with the more general SMDP "options" framework?
2.  How sensitive is the mask inference to the quality of the single demonstration? What happens if the demo is suboptimal (e.g., takes a long but feasible path) or noisy (e.g., contains a skill that is, in fact, infeasible)?
3.  Can you report the test-time sampling efficiency? Specifically, what is the average rejection rate (or number of samples) in Algorithm 2 for the different target environments? One might expect the highly-constrained Maze2D tasks to have a high rejection rate.

---

> ### Author Response · Authors · 2025-11-22
> **Official Comment by Authors (1/2)**
>
> Thank you very much for your insightful and constructive feedback on our submission. The following are our detailed responses.
>
> ---
>
> ### Weakness 1:
>
> > "Misleading Presentation of "Options": The paper's framing around the "options" framework is a bit oversell."
>
> ### Question 1:
>
> > "Given that the theory and implementation rely on fixed-horizon skills $H$, would the authors be willing to clarify this in the main text (e.g., by framing it as an "augmented action-space MDP") to avoid confusion with the more general SMDP "options" framework?"
>
> **Response**
>
> We thank the reviewer for this insightful comment. We agree that our method does not fully address the general SMDP framework due to the fixed skill horizon assumption. To clarify this, we have revised the main text to consistently refer to our framework as an "augmented action-space MDP" and refer to skills as a simplified version of "options" throughout the paper. We believe these changes improve the clarity and precision of our contribution.
>
> ---
>
> ### Weakness 2:
>
> > "Simplistic Mask Inference: The test-time mask inference is a simple "set $m(z)=1$ if $z$ is in the demo." This assumes the single demonstration is both correct and comprehensive. The paper seems doesn't explore how robust this is to suboptimal or noisy demonstrations. And a soft version seems to be more plausible than hard assignment for real world application."
>
> ### Question 2:
>
> > "How sensitive is the mask inference to the quality of the single demonstration? What happens if the demo is suboptimal (e.g., takes a long but feasible path) or noisy (e.g., contains a skill that is, in fact, infeasible)?"
>
> **Response**
>
> We agree with the reviewer that MSTT indeed assumes that (1) target demonstration can accomplish the task, and (2) any skill observed in the single demonstration is feasible and available in the source dataset. We would like to claim it is a feature of MSTT that it values the **safety requirement induced by the demonstration**.
>
> 1. Suboptimal skill: MSTT considers any skill that is not present in the demonstration to be unsafe to use, as using it may cause damage. Therefore, MSTT is designed to safely adapt to the target domain. For suboptimal but feasible demonstration, MSTT is able to extract the feasible skills and combine them to find a potentially more efficient path. For example as shown in `Figure 15` in the appendix (`Section D.3`), although the demonstration detours first, MSTT is still able to find the optimal path based on the pre-trained value function.
>
> 2. Infeasible skill: if the demonstration has new skills that are not present in the source dataset, then inferring the mask from it will lead to random behavior. MSTT works in an offline manner, and it does not have online exploration. If a user wants the agent to achieve a task in the target domain that requires new skills, additional data collection and skill learning would be necessary. Enabling continual learning on new skill trajectory could potentially help discover the optimal solution in such cases, which is an interesting direction, and we would like to explore as future work.
>
> Extending MSTT to have soft version of the mask inference is indeed an interesting direction. One potential approach could involve updating the mask transition matrix in a Bayesian manner and use the posterior after observing the demonstration, allowing the model to capture uncertainty in skill feasibility. We will explore this as future work.
>
> ---

---

> > ### Author Response · Authors · 2025-11-22
> > **Official Comment by Authors (2/2)**
> >
> > ### Weakness 3:
> >
> > > "The test-time adaptation (Algorithm 2) is a rejection sampler. Potential Inefficiency: In highly constrained target environments (where most $m(z)=0$), this could be extremely inefficient, requiring many samples from the diffusion model to find one valid option."
> >
> > ### Question 3:
> >
> > > "Can you report the test-time sampling efficiency? Specifically, what is the average rejection rate (or number of samples) in Algorithm 2 for the different target environments?"
> >
> > **Response**
> >
> > MSTT's diffusion based policy samples skills by conditioning on the current state, which helps improve the sampling efficiency compared to state independent sampling. During test time, we select the best feasible skill among 32 skill samples generated from the diffusion model at each decision step. We find that when the current state is in the feasible skill region, the average acceptance rate is 45.9\%±3.1\% across all three target environments in Maze2D.
> >
> > ---
> >
> > ### Weakness 4:
> >
> > > "Closely related work."
> >
> > **Response**
> >
> > We appreciate the reviewer bringing these relevant works to our attention. We have incorporated a discussion of them into the related work section of the revised manuscript. These work are closely related in that they also model temporally extended skills/options as abstract actions in an MDP and provide theoretical guarantees via MDP homomorphisms. The masked Bellman operator in MSTT can be potentially applied on top of these abstract models to incorporate more general options. We would like to extend our work to skills and options with different horizons as future work, potentially building upon the frameworks proposed in these papers.
> >
> > ---
> >
> > We sincerely thank you for your encouraging evaluation and thoughtful comments. Your recognition of our work’s contributions is highly motivating, and your suggestions have helped us further strengthen the clarity and rigor of the manuscript.

---

> ### Comment · Reviewer_DAdX · 2025-11-24
> **score raised as concerns addressed**
>
> We thank authors for their effort and raise scores accordingly.

---

> > ### Author Response · Authors · 2025-11-24
> >
> > Dear Reviewer DAdX,
> >
> > Thank you for your kind feedback and for taking the time to review our revision. We are grateful for your positive evaluation and appreciate your support in raising the score. Your insightful comments have helped us further improve our work.
> >
> > Best regards,
> >
> > The Authors

---

### Official Review · Reviewer_J5hB · 2025-10-31

**Soundness:** 2
**Presentation:** 2
**Contribution:** 2
**Rating:** 4
**Confidence:** 2

**Summary:**

The paper proposes Masked Skill Token Training (MSTT), an offline hierarchical reinforcement learning framework for off-dynamics transfer using only observation-only demonstrations. MSTT learns a discrete skill space via VQ-VAE and trains a feasibility-conditioned critic through masked Bellman updates. At deployment, it infers a binary skill mask from a single demonstration and uses a diffusion-based trajectory generator to execute only feasible skills. Experiments on Maze2D, FetchReach, and Habitat environments show that MSTT achieves robust off-dynamics transfer.

**Strengths:**

This paper addresses the important problem of transferring policies using only observation-based demonstrations. By introducing a simple masking mechanism over discretized skills, the method enables efficient offline adaptation. The proposed framework demonstrates strong generalization and robustness for off-dynamics transfer, and further presents experiments combining MSTT with VLMs, showing its potential for low-effort policy adaptation.

**Weaknesses:**

1. The masking mechanism relies on random sampling during training, which seems inefficient. Could the masking process be learned? Moreover, MSTT's scalability is limited as the skill vocabulary grows, might cause the critic to overfit to partial masking patterns and weaken generalization.

2. The proposed method mainly addresses structural dynamics shifts (e.g., blocked transitions) and does not handle continuous or parametric changes in dynamics (e.g., friction, mass, damping). This narrows its applicability.

3. Visualizing which discretized skills become masked in each target domain could provide strong qualitative and interpretable evidence of proper adaptation. By examining how feasibility patterns change across domains, the authors could further demonstrate the effectiveness of the proposed method.

4. The baselines, such as BC and Diffuser, are not specifically designed for off-dynamics transfer, limiting the fairness of comparison. It would strengthen the evaluation to include methods such as SiMPL[1], PDT[2], MetaDiffuser[3], HDP[4], which explicitly target policy generalization or meta-adaptation across dynamics shifts. Furthermore, the hierarchical skill-based approaches are missing.

    [1] Skill-based Meta-Reinforcement Learning, ICLR 2022

    [2] Prompting Decision Transformer for Few-Shot Policy Generalization, ICML 2022

    [3] MetaDiffuser: Diffusion Model as Conditional Planner for Offline Meta-RL, ICML 2023

    [4] Hyper-Decision Transformer for Efficient Online Policy Adaptation, ICLR 2023

**Questions:**

1. In the VLM + MSTT case study, does the VLM function as a world model that generates visual observation trajectories to serve as demonstration inputs for the policy transfer?

---

> ### Author Response · Authors · 2025-11-22
> **Official Comment by Authors (1/2)**
>
> Thank you very much for your insightful and constructive feedback on our submission. The following are our detailed responses.
>
> ---
>
> ### Weakness 1:
>
> > "The masking mechanism relies on random sampling during training, which seems inefficient. Could the masking process be learned? The critic might overfit to partial masking patterns and weaken generalization."
>
> **Response**:
>
> We thank the reviewer for this insightful comment. We agree that learning the masking process can potentially further improve the efficiency of our method.
>
> 1. **Learning masking process**: To evaluate the possibility of learning the masking process, we have conducted experiments where the masking process is dataset dependent by following similar distributions of skills as the source dataset. We label all the skills (short trajectories) in the source dataset with their corresponding skill token indices, which leads to sequences of skill indices and an empirical transition matrix estimated by counting the occurrences of skill transitions. A mask can be generated by labeling the available skills sampled from the transition matrix with certain steps. As shown in **Table 1** below, MSTT with learned masking achieves reasonable performance but is still outperformed by random masking. We hypothesize that this is because the random masking strategy provides sufficient coverage of the skill space during training, allowing the critic to generalize well at test time.
>
> | Table 1 (Reviewer-J5hB) | | Env T1 | | | Env T2 | | | Env T3 | |
> |---|:---:|:---:|:---:|:---:|:---:|:---:|:---:|:---:|:---:|
> | | Return ($\uparrow$) | Steps ($\downarrow$) | Goal ($\uparrow$) | Return ($\uparrow$) | Steps ($\downarrow$) | Goal ($\uparrow$) | Return ($\uparrow$) | Steps ($\downarrow$) | Goal ($\uparrow$) |
> | Random Masking | 163.97 | 186.92 | 89% | 145.51 | 211.01 | 95% | 111.59 | 249.33 | 99% |
> | Learned Masking | 154.68 | 196.10 | 78% | 134.56 | 216.28 | 84% | 82.93 | 267.91 | 84% |
>
> 2. **Overfitting to partial masking patterns**: While the skill library can grow large with a bigger codebook size, we find that the skills learned by VQ-VAE are often well-structured and the transitions between skills are sparse. `Figure 11` in the appendix (`Section D.2`) illustrates the transition matrix of the skill tokens extracted from the source dataset. The clear banded structure indicates that only a small subset of skills is likely to transition to each other. This property helps mitigate the risk of overfitting to partial masking patterns, as many masking patterns become irrelevant due to the sparse transitions.<br>
> We evaluate the generalization over masks by validating on a fixed set of masks that are not used for sampling during training. `Section D.2` in the appendix presents the results shown in `Figure 12`, where the performance on the validation masks set is comparable to that on the training masks, indicating good generalization.
>
> These results and our prior experiments demonstrate that random mask sampling provides a simple yet effective way to expose the critic to a diverse set of masking patterns. For larger number of skills, one can learn an autoregressive model over the skill indices on the datasets and sample masks from the model. We view large-scale codebook learning and mask sampling strategy as an interesting direction and believe that our work provides insightful findings for this direction.
>
> ---
>
> ### Weakness 2:
>
> > "The proposed method mainly addresses structural dynamics shifts (e.g., blocked transitions) and does not handle continuous or parametric changes in dynamics (e.g., friction, mass, damping)."
>
> **Response**:
>
> We thank the reviewer for this insightful comment. MSTT is primarily designed to handle structural dynamics changes that lead to binary feasibility of skills. However, we believe that MSTT can be extended to address certain types of continuous dynamics shifts. To this end, we create a hybrid dataset that combines both structural and continuous dynamics changes, where the trajectories are collected with the mass of the agent being 0.1x, 1x or 10x of the original mass.
>
> | Table 2 (Reviewer-J5hB) | | Env T1 | | | Env T2 | | | Env T3 | |
> |---|:---:|:---:|:---:|:---:|:---:|:---:|:---:|:---:|:---:|
> | | Return ($\uparrow$) | Steps ($\downarrow$) | Goal ($\uparrow$) | Return ($\uparrow$) | Steps ($\downarrow$) | Goal ($\uparrow$) | Return ($\uparrow$) | Steps ($\downarrow$) | Goal ($\uparrow$) |
> | Hybrid Dataset | 159.83 | 191.15 | 88% | 142.94 | 208.04 | 88% | 84.45 | 266.45 | 90% |
>
> **Table 2** summarizes the results, showing that MSTT can still effectively adapt to the dynamics shifts by inferring the skills from the target demonstration. We would like to explore this direction further in future work by extending the dynamics shift to more fine-grained changes with larger codebook sizes.
>
> ---

---

> > ### Author Response · Authors · 2025-11-22
> > **Official Comment by Authors (2/2)**
> >
> > ### Weakness 3:
> >
> > > "Visualizing which discretized skills become masked in each target domain could provide strong qualitative and interpretable evidence of proper adaptation."
> >
> > **Response**:
> >
> > To address this, we have added visualizations of all the skills and unmasked skills in different target domains in the revised manuscript (`Section D.1`). `Figure 9` illustrates the skills decoded from the pre-trained VQ-VAE model, demonstrating its semantic meanings. `Figure 10, 11, 12` show the skills available under the inferred mask from the target demonstration. They correspond well with the environmental changes. These visualizations provide qualitative and interpretable evidence that MSTT effectively identifies and utilizes feasible skills in the target domains.
> >
> > ---
> >
> > ### Weakness 4:
> >
> > > "It would strengthen the evaluation to include methods such as SiMPL[1], PDT[2], MetaDiffuser[3], HDP[4], which explicitly target policy generalization or meta-adaptation across dynamics shifts."
> >
> > **Response**:
> >
> > We appreciate the reviewer bringing these relevant works to our attention. Following your suggestion, we have added additional experimental evaluations and incorporated a discussion of them into the related work section of the revised manuscript.
> >
> > SiMPL and MetaDiffuser require multiple reward functions to meta-learn the skill policy or task context vector. PDT and HDP learn in a meta-learning manner to encode the task, which requires multiple source tasks' expert demonstrations to learn the task characteristics. MSTT differs from them in that it discovers potential dynamic shifts from a single offline dataset without expert demonstrations from multiple source tasks. While comparing with them would be ideal, they cannot be directly trained in our datasets, and it would require designing multiple tasks and collecting additional expert data. To the best of our knowledge, no existing method can achieve observation only transfer from a single source dataset. We selected the baselines because they represent the closest existing approaches to our problem setting.
> >
> > To address your concern over the baselines, we conducted a comparison with an offline off-dynamics RL method IGDF as suggested by reviewer Z4mu. The results in **Table 3** below demonstrated the effectiveness of MSTT.
> >
> > | Table 3 (Reviewer-J5hB) | | Env T1 | | | Env T2 | | | Env T3 | |
> > |---|:---:|:---:|:---:|:---:|:---:|:---:|:---:|:---:|:---:|
> > | | Return ($\uparrow$) | Steps ($\downarrow$) | Goal ($\uparrow$) | Return ($\uparrow$) | Steps ($\downarrow$) | Goal ($\uparrow$) | Return ($\uparrow$) | Steps ($\downarrow$) | Goal ($\uparrow$) |
> > | IGDF | 27.27 | 204.35 | 81% | 23.54 | 260.05 | 95% | 53.82 | 212.65 | 97% |
> > | MSTT | 163.97 | 186.92 | 89% | 145.51 | 211.01 | 95% | 111.59 | 249.33 | 99% |
> >
> > ---
> >
> >
> > ### Question 1:
> >
> > > "In the VLM + MSTT case study, does the VLM function as a world model that generates visual observation trajectories to serve as demonstration inputs for the policy transfer?"
> >
> > **Response**:
> >
> > We agree that in these cases the VLM serves as a world model that generates observational trajectories based on the provided high-level instructions. These generated trajectories act as the demonstration inputs for MSTT to infer the skill mask, select skills and predict actions to accomplish the task.
> >
> > ---
> >
> > We truly appreciate the care and clarity in your feedback. We hope the additional analyses and comparison have fully addressed your remaining concerns. We’ve genuinely learned from your perspective and would be glad to discuss any points that may remain.

---

> > > ### Comment · Reviewer_J5hB · 2025-11-24
> > >
> > > I appreciate the authors for their detailed response. My concerns have been resolved, and I have increased my rating accordingly.

---

> ### Author Response · Authors · 2025-11-24
>
> Dear Reviewer J5hB,
>
> Thank you very much for your time and effort in reviewing our response and revised manuscript. We are grateful that the revisions addressed your concerns, and we sincerely appreciate your support in raising the rating. Your insightful comments have been invaluable in helping us further improve our work.
>
> Best regards,
>
> The Authors

---

### Official Review · Reviewer_Z4mu · 2025-11-01

**Soundness:** 2
**Presentation:** 2
**Contribution:** 2
**Rating:** 6
**Confidence:** 2

**Summary:**

This paper proposes MSTT, an offline hierarchical learning method for off-dynamics transfer. It treats the dynamics discrepancy as blocking some learnt skills and thus masking those out  at deployment. The target skill mask is inferred based on single observation-only demonstration which makes it suitable for domains where action annotations or environment interactions are costly. Empirical results show MSTT shows higher return and less complete steps than DARC and BC in Maze2d transfer.

**Strengths:**

1. The method proposed is novel to me. Modeling dynamics discrepancy as masked out skill tokens alleviate the problem presented in specific transfer learning problems.

2. Proposed MSTT shows better performance in the demonstration-only offline transfer learning setting over some imitation and offline learning baselines.

**Weaknesses:**

1. The assumption that source and target domains only differ in feasibility distinction of specific skills is strong to me. This limits the generalizability of the method proposed, and raises my concern to the setting proposed that target dataset should have similar levels of accessibility compared to source dataset.

2. Weak baseline comparison. For cross-domain imitation learning, [1] suggests a method which also learns from cross domain state-only demonstrations, which explicitly deals with domain discrepancy upon ID methods. Moreover, as a minor concern, DARA is not a recent offline transfer method though it doesn't share the same setting with MSTT. I would suggest replace it with some stronger baselines such as [2].

[1] Cross-domain Imitation from Observations Dripta S. Raychaudhuri, Sujoy Paul, Jeroen Vanbaar, Amit K. Roy-Chowdhury Proceedings of the 38th International Conference on Machine Learning, PMLR 139:8902-8912, 2021.

[2] Contrastive Representation for Data Filtering in Cross-Domain Offline Reinforcement Learning Xiaoyu Wen, Chenjia Bai, Kang Xu, Xudong Yu, Yang Zhang, Xuelong Li, Zhen Wang Proceedings of the 41st International Conference on Machine Learning, PMLR 235:52720-52743, 2024.

**Questions:**

1. Related to W1, is it possible that dynamics discrepancy leads to new skills that have not been learnt with source domain dataset, in which case the application of MSTT may lead to suboptimal performance?

2. How does the codebook size affect the downstream performance?

---

> ### Author Response · Authors · 2025-11-22
> **Official Comment by Authors (1/2)**
>
> Thank you very much for your insightful and constructive feedback on our submission. The following are our detailed responses.
>
> ---
>
> ### Weakness 1:
>
> > "The assumption that source and target domains only differ in feasibility distinction of specific skills is strong to me. This limits the generalizability of the method proposed, and raises my concern to the setting proposed that target dataset should have similar levels of accessibility compared to source dataset."
>
> ### Question 1:
>
> > "Related to W1, is it possible that dynamics discrepancy leads to new skills that have not been learnt with source domain dataset, in which case the application of MSTT may lead to suboptimal performance?"
>
> **Response**:
>
> We agree with the reviewer that MSTT indeed assumes skills required to complete the target task are a subset of source dataset. This applies to many practical scenarios where the target environment imposes additional constraints (e.g., obstacles, restricted areas) that limit the agent's ability to utilize certain skills learned in the source domain. MSTT works in an offline manner, and it does not require online exploration through interaction or actions associated with the observations in the target demonstration. It is a feature of MSTT that it values the **safety requirement induced by the demonstration**. Any skill that is not present in the demonstration is considered unsafe to use, as using it may cause damage. Therefore, MSTT is designed to safely adapt to the target domain. It is worth noting that MSTT has the ability in combining the skills learned from the source domain in novel ways to achieve tasks never demonstrated before, as long as the individual skills are feasible. If the target environment enables unknown state region to accomplish the task more efficiently, MSTT may not discover it due to its offline nature.
>
> If a user wants the agent to achieve a task in the target domain that requires new skills, additional data collection and skill learning would be necessary. Enabling continual learning on new skill trajectory could potentially help discover the optimal solution in such cases, which is an interesting direction, and we would like to explore as future work.
>
> ---
>
> ### Weakness 2:
>
> > Weak baseline comparison.
>
> **Response**:
>
> We appreciate your suggestion for considering xDIO [1] and IGDF [2] as baselines.  xDIO targets translating expert demonstrations from a source domain to a target domain with different morphologies. IGDF presents a cross-domain offline RL method that addresses dynamics shifts by filtering source domain data using a contrastive cross-domain representation learning approach, which requires access to the actions and rewards associated with the states in target domain. We have incorporated a discussion of them into the related work section of the revised manuscript.
>
> Following your suggestion, we have added IGDF as an additional baseline in our experiments. While xDIO also focuses on learning from observations, it targets a different problem setting of morphology mismatch, and aims to copy the expert skill of the source robot to the target robot under the same task, which is different from our setting of dynamics mismatch.
>
> | Table 1 (Reviewer-Z4mu) | | Env T1 | | | Env T2 | | | Env T3 | |
> |---|:---:|:---:|:---:|:---:|:---:|:---:|:---:|:---:|:---:|
> | | Return ($\uparrow$) | Steps ($\downarrow$) | Goal ($\uparrow$) | Return ($\uparrow$) | Steps ($\downarrow$) | Goal ($\uparrow$) | Return ($\uparrow$) | Steps ($\downarrow$) | Goal ($\uparrow$) |
> | IGDF | 27.27 | 204.35 | 81% | 23.54 | 260.05 | 95% | 53.82 | 212.65 | 97% |
> | MSTT | 163.97 | 186.92 | 89% | 145.51 | 211.01 | 95% | 111.59 | 249.33 | 99% |
>
> We have conducted experiments on the Maze2D environment, and the results are summarized in **Table 1**. To adapt IGDF to our setting, we assume that the target demonstration has additional information of actions and per-step rewards associated with the observations. IGDF originally assumes the access to hundreds of thousands of transitions in each domain to learn the cross-domain representation. Although better than DARA, it still under-performs MSTT in our setting with only a single demonstration in the target domain.
>
> [1] Cross-domain Imitation from Observations Dripta S. Raychaudhuri, Sujoy Paul, Jeroen Vanbaar, Amit K. Roy-Chowdhury Proceedings of the 38th International Conference on Machine Learning, PMLR 139:8902-8912, 2021.
>
> [2] Contrastive Representation for Data Filtering in Cross-Domain Offline Reinforcement Learning Xiaoyu Wen, Chenjia Bai, Kang Xu, Xudong Yu, Yang Zhang, Xuelong Li, Zhen Wang Proceedings of the 41st International Conference on Machine Learning, PMLR 235:52720-52743, 2024.
>
> ---

---

> > ### Author Response · Authors · 2025-11-22
> > **Official Comment by Authors (2/2)**
> >
> > ### Question 2:
> >
> > > "How does the codebook size affect the downstream performance?"
> >
> > **Response**:
> >
> > The codebook size determines the granularity of the skills learned by the VQ-VAE. A smaller codebook size leads to coarser skills that are less expressive, while a larger codebook size allows for a more diverse set of skills, which can potentially improve performance by providing finer control over the agent's behavior. However, it also increases the complexity of the skill space, which may make it more challenging for the critic to learn effective value estimates.
> >
> > We have conducted experiments with different codebook sizes (e.g., 15, 64, 240, 512 and 1000) and **Table 2** below summarizes the results on Maze2D environment. We observe that performance generally improves with larger codebook sizes, and the benefits plateau after certain points.
> >
> > | Table 2 (Reviewer-Z4mu) | | Env T1 | | | Env T2 | | | Env T3 | |
> > |---|:---:|:---:|:---:|:---:|:---:|:---:|:---:|:---:|:---:|
> > | Codebook Size | Return ($\uparrow$) | Steps ($\downarrow$) | Goal ($\uparrow$) | Return ($\uparrow$) | Steps ($\downarrow$) | Goal ($\uparrow$) | Return ($\uparrow$) | Steps ($\downarrow$) | Goal ($\uparrow$) |
> > | 15 | 204.57 | 146.43 | 100% | 0.0 | 350.0 | 0% | 84.32 | 266.28 | 60% |
> > | 64 | 120.78 | 230.1 | 88% | 23.06 | 327.24 | 30% | 0.82 | 349.19 | 1% |
> > | 240 | 163.97 | 186.92 | 89% | 145.51 | 211.01 | 95% | 111.59 | 249.33 | 99% |
> > | 512 | 191.04 | 159.9 | 94% | 135.39 | 215.51 | 90% | 124.19 | 226.8 | 99% |
> > | 1000 | 206.36 | 144.64 | 100% | 168.31 | 182.66 | 97% | 92.04 | 258.82 | 86% |
> >
> > ---
> >
> > We truly appreciate the care and clarity in your feedback. Your recognition of our work’s contributions is highly motivating, and your suggestions have helped us further strengthen the clarity and rigor of the manuscript.

---

> > > ### Comment · Reviewer_Z4mu · 2025-11-26
> > >
> > > I thank the reviewers for their detailed comments and additional experiments. The response has clarified my concerns regarding the baseline and hyperparameter sensitivity and I will maintain my current positive rating.

---

> > > > ### Author Response · Authors · 2025-11-26
> > > >
> > > > Dear Reviewer Z4mu,
> > > >
> > > > Thank you for taking the time to review our revision. We appreciate your positive rating and are glad that our responses addressed your concerns. Your insightful comments have been invaluable in helping us further improve our work.
> > > >
> > > > Best regards,
> > > >
> > > > The Authors

---

### Author Response · Authors · 2025-12-03
**Summary of Discussions and Revisions**

Dear Reviewers, ACs, SACs, and PCs,

We would like to express our sincere gratitude for your support, and deeply appreciate the time and care you continue to invest in the review process. We thank all reviewers' constructive suggestions and detailed questions, all of which have directly contributed to strengthening and clarifying our manuscript. Furthermore, after our response, reviewers (`Z4mu`, `J5hB` and `DAdX`) acknowledged that our responses have resolved their concerns. In particular, reviewer `J5hB` has raised rating from 4 to 6, and reviewer `DAdX` has raised rating from 6 to 8. We have prepared a concise summary of our earlier discussions, outlining our key contributions and how we addressed each concern. We hope this will be helpful:

**Across the reviews, we are glad to receive several consistent points of positive feedback for our contributions**:

- **Novel and elegant idea**: Reviewers consistently acknowledged that our method is novel (`Z4mu`, `DAdX`, `VHqp`). Reviewers praised that the ideas of masked Bellman operator and sampling skill masks for simulating dynamics mismatch are elegant (`DAdX`, `VHqp`).

- **Efficient transfer**: `Z4mu` and `J5hB` both highlighted the efficiency of our method in transferring policies to new dynamics in an offline manner. Our proposed skill masking for modeling dynamics discrepancies alleviates the requirement of environments with multiple different dynamics.

- **Sound theoretical foundation**: Our method of masked Bellman operator is justified by novel theoretical analysis, which links the approximation error directly to skill determinism and provides a principled understanding of dynamics shift modeling (`DAdX`).

- **Practical problem formulation**: The targeted problem of adapting offline-trained agents to new environmental dynamics using only observation-only demonstrations without costly interaction or action labels is important (`J5hB`) and practical (`DAdX`), which addresses a critical challenge of generalization in RL (`VHqp`). Reviewers all recognized the successful experimental performance in achieving strong generalization and robustness for off-dynamics transfer.

**In direct response to the reviewers’ feedback, we have**:

- **Demonstrated masking semantics and efficiency**: We provided evidence (`Fig. 13 & 14`, `Section D.2` in appendix) of the effectiveness of random masking thanks to the semantically meaningful skills and sparsity of skill transitions (`J5hB`, `VHqp`). We conducted experiments (Table 1 (Reviewer-J5hB)) comparing against learned masking strategies based on skill transition with results demonstrating random masking provides better regularization and generalization (`J5hB`, `VHqp`). We showed the sampling efficiency of skills with conditioning factor and acceptance rate (`DAdX`).

- **Addressed demonstration robustness**: We clarified that our mask inference is designed as a safety feature for offline adaptation (`Z4mu`). We demonstrated through examples (`Fig. 15 & 16`, `Section D.3`) how the method handles suboptimal demonstrations by combining feasible skills to find potentially more efficient paths, while avoiding unsafe skills (`DAdX`, `VHqp`). This safe approach is appropriate for real-world deployment.

- **Extended to continuous dynamics shifts**: We extended our experiments to evaluate on both structural and continuous dynamics changes. Results on a hybrid dataset (Table 2 (Reviewer-J5hB)) demonstrate the ability of our method in handling some continuous shifts, and open valuable avenues for future research to more fine-grained changes (`J5hB`, `VHqp`).

- **Added related work and baseline comparison**: Following suggestions, we added additional baseline (Table 1 (Reviewer-Z4mu)) and our method outperforms the offline off-dynamics RL method IGDF (`Z4mu`). We incorporated discussions of related work including observation based imitation (xDIO), meta-RL methods (SiMPL, PDT, MetaDiffuser, HDP), and hierarchical option learning (HiT-MDP, VMOC) with clearer terminology (`Z4mu`, `J5hB`, `DAdX`), acknowledging their relevance and noting the potential of improvement based on them as interesting future work.

- **Hyperparameter analysis and qualitative visualizations**: We conducted ablations across codebook sizes and showed performance (Table 2 (Reviewer-Z4mu)) generally improves with larger codebooks while plateauing after certain points (`Z4mu`, `VHqp`). Our added visualizations (figures in `Section D.1`) of learned skills and unmasked skills in target domains provide qualitative evidence of semantic meaningfulness and proper adaptation (`J5hB`, `VHqp`).

Once again, we thank all reviewers for their valuable feedback. We believe our revisions have addressed all concerns, as detailed in the responses below. We appreciate the AC’s effort in managing the review.

Best regards,

The Authors

---

### Meta-Review · Area_Chair_f2e4 · 2026-01-07

**Summary:**

The paper introduces Masked Skill Token Training (MSTT), a hierarchical offline RL framework designed to transfer policies to environments with altered dynamics using only observation-based demonstrations. The reviews were positive. Reviewers praised the "masked Bellman operator" as an elegant and novel mechanism for simulating dynamics shifts without requiring multi-environment data. The practical setting and promising results on multiple benchmarks were highlighted as key strengths. In the rebuttal, the authors have addressed concerns about baseline strength, masking strategies, and robustness to continuous dynamics, leading two reviewers to raise their scores.

One minor issue to note is a bibliographic error in the references: the citation to Sutton and Barto’s *Reinforcement Learning: An Introduction* lists an incorrect venue. This is a well-known textbook published by MIT Press rather than an IEEE journal, and the authors should correct this in the final version.

**Reviewer Concerns:**

- Reviewers Z4mu and J5hB initially requested stronger baselines. The authors added a comparison to IGDF. They also expanded the related work discussion to include relevant meta-RL and imitation learning methods.

- Concerns about the efficiency of random masking were resolved by experiments showing that random masking actually outperforms learned masking strategies.

- Reviewers J5hB and VHqp asked about continuous shifts. The authors provided new results on a hybrid dataset with mass variations, demonstrating robustness to some continuous parametric changes.

- Reviewer DAdX pointed out that the term "options" was oversold given the fixed-horizon skills. The authors agreed to clarify the formulation as an "augmented action-space MDP."

- The reliance on a single demonstration to infer the mask remains a fundamental constraint. If the demonstration is noisy or fails to cover necessary skills, the method may struggle.

Most of these concerns have been addressed. For the last concern, the authors acknowledged this is inherent to the offline setting but showed robustness to suboptimaldemonstrations.

**Reviewer Scores:**

Two reviewers raising their ratings following the rebuttal. Reviewer DAdX increased their score from 6 to 8 and Reviewer J5hB raised their score from 4 to 6. Reviewers Z4mu and VHqp maintained their positive ratings of 6

---

### Decision · Program_Chairs · 2026-01-26

Accept (Poster)